# Problem-solving skills are predicted by technical innovations in the wild and brain size in passerines

Jean-Nicolas Audet ● [1,2,3] ✉, Mélanie Couture ● [1,4], Louis Lefebvre ● [5,6] &
Erich D. Jarvis ● [1,2,3,4]

Behavioural innovations can provide key advantages for animals in the wild, especially when ecological conditions change rapidly and unexpectedly. Innovation rates can be compared across taxa by compiling field reports of novel behaviours. Large-scale analyses have shown that innovativeness reduces extinction risk, increases colonization success and is associated with increased brain size and pallial neuron numbers. However, appropriate laboratory measurements of innovativeness, necessary to conduct targeted experimental studies, have not been clearly established, despite decades of speculation on the most suitable assay. Here we implemented a battery of cognitive tasks on 203 birds of 15 passerine species and tested for relationships at the interspecific and intraspecific levels with ecological metrics of innovation and brain size. We found that species better at solving extractive foraging problems had higher technical innovation rates in the wild and larger brains. By contrast, performance on other cognitive tasks often subsumed under the term behavioural flexibility, namely, associative and reversal learning, as well as self-control, were not related to problem-solving, innovation in the wild or brain size. Our study yields robust support for problem-solving as an accurate experimental proxy of innovation and suggests that novel motor solutions are more important than self-control or learning of modified cues in generating technical innovations in the wild.

Animals vary in their likelihood of inventing new behavioural solutions to ecological problems. Two approaches have been used to understand this variation: large-scale analyses of ecological, evolutionary and neural correlates of observational data taken from the wild[1–7], and experimental studies on smaller numbers of species[8–11]. Integrating these approaches is crucial to obtain a more comprehensive and generalizable understanding of cognition. Innovations in the wild, as these novel

solutions were first called by Kummer and Goodall[12], are thought to involve three cognitive components: (1) inhibition of habitual responses when an animal realizes they do not work, (2) exploration of new actions to solve the problem and (3) learning of modified cues associated with the solution. Each component can be targeted experimentally using specific behavioural assays: (1) self-control tasks, which measure the ability to inhibit a prepotent but unproductive behaviour (for example,

[1]The Rockefeller University Field Research Center, Millbrook, NY, USA. [2]Howard Hughes Medical Institute, Chevy Chase, MD, USA. [3]Laboratory of Neurogenetics of Language, The Rockefeller University, New York, NY, USA. [4]The Vertebrate Genome Laboratory, The Rockefeller University, New York, NY, USA. [5]Department of Biology, McGill University, Montreal, Quebec, Canada. [6]CREAF, Autonomous University of Barcelona, Cerdanyola del Vallès, Spain. ✉e-mail: jaudet@rockefeller.edu

**Fig. 1 | Links between field measures of innovations, brain size and their potential laboratory measurements, assessed in this study. a**, Foraging innovations, as well as dietary generalism, have been well documented in the field. Yet, the cognitive skills responsible for innovations are poorly understood, and their appropriate experimental measurement has not been clearly identified. **b**, Brain size has been shown to be associated with innovativeness and dietary generalism in the wild, but it is unclear how brain size varies with different experimental assays of cognition or their covariates across species. **c**, The most common experimental tasks assumed to be linked with innovativeness include problem-solving, associative learning, reversal learning and self-control. However, a link between performance on these tasks, their covariates and innovativeness across species has yet to be shown. All the traits measured by these assays, as well as innovativeness, are often considered components of behavioural flexibility. Solid black arrows show known relationships, and dashed grey arrows show untested or equivocal relationships across species. Image credits: Louis Lefebvre for the Barbados bullfinch in **a**, Christopher Torres (University of Texas at Austin) for the brain endocast in **b** and Jean-Nicolas Audet for the nuthatch solving a problem in captivity in **c**.

finding an alternative way to obtain a reward), assess inhibitory control[13]; (2) puzzle boxes requiring extractive foraging and obstacle removal, sometimes with tools, assess novel problem-solving[14]; and (3) association and reversal tasks, which measure the ability to discriminate between rewarded and unrewarded cues, target the efficiency of learning new cues[15]. All these assays have been considered measures of behavioural flexibility, the ability to adjust a behaviour in response to changing conditions[16], for example, when colonizing new areas such as cities. Consistent with this notion, problem-solving speed is positively associated with the degree of urbanization (for example, refs. 10,17) and consumption of anthropogenic food[18]. By contrast, associative and reversal learning speed has been found to correlate negatively with urbanization[19,20]. A concept such as behavioural flexibility can only subsume different traits under a common denominator if their measures produce concordant patterns. Dozens of studies have been conducted on birds and yielded unclear results, partly because the vast majority have focused on a small number of species, most often one (for example, ref. 18), and/or only one or two of the three types of assay (for example, ref. 21). The correspondence between field measures of innovation and experimental assays of flexibility is thus an open question (Fig. 1).

In this Article, we addressed these issues with a large sample of species and variables, testing interspecific and interindividual relationships between self-control, associative learning, reversal learning and problem-solving on 203 individuals from 15 passerine species, including 13 wild-caught and 2 domesticated species (Fig. 2a and Supplementary Table 1). We tested mainly male birds to reduce sex as a variable, except for 7 female birds in two species where it was difficult to obtain sufficient wild-caught sample sizes of male birds (Methods). We explored associations between performance on experimental assays of cognition and with measures of absolute and allometrically corrected brain size, innovation rates in the wild and several potential covariates; in a separate study[22], we also explored associations with vocal learning complexity. If the assays are all valid measures of behavioural flexibility, then we expect them all to be positively associated with each other as well as with innovation rate in the wild and absolute and allometrically corrected brain size (Fig. 2b,c). If, as some data suggest (reviewed in ref. 16), however, behavioural flexibility is a heterogeneous concept, not all assays will be correlated. In particular, persistence is one of the variables that favour solving an extractive

foraging problem (for example, refs. 9,17,23,24), but it is also one of the primary sources of error in reversal learning[25], which would lead to negative or non-significant relationships between assays, as well as with innovation and brain size (Fig. 2d).

## Results

### Behavioural assays measure distinct skills

We first compared performance on all cognitive assays for 12 to 19 individuals from each of the 15 passerine species. Passerines represent half of the world's avian species. Our sample includes representatives of known high (for example, Corvidae, Turdidae, Sturnidae) and low (for example, Estrildidae, Passerellidae, Troglodytidae) innovation families that together account for 967 cases of innovation (out of 4,455) in ref. 26. To measure problem-solving, we used four different obstacle-removal tasks, each requiring a different motor pattern to obtain a food reward: pulling/knocking, flipping/grabbing, piercing/tearing or dragging a moveable element of the apparatus to extract the food (Extended Data Fig. 1f–i and Supplementary Videos 1–4). Performance was significantly associated across all four problem-solving tasks between the 15 species (Extended Data Fig. 2 and Supplementary Table 2a). Therefore, we used the average number of trials to solve the four tasks as our measure of problem-solving performance. To assess self-control, we used a detour-reaching paradigm in which the birds had to access a food reward from the open ends of a cylinder containing food visible behind a transparent barrier (Extended Data Fig. 1j and Supplementary Videos 5 and 6); a bird's initial response to a task such as this is to approach the part where the food is visible behind the transparent barrier and peck at it; to be successful, the animal has to inhibit this first response and move away to the open end of the apparatus without pecking at the barrier. Associative (acquisition) learning was assessed by giving the birds a two-colour discrimination task, where they had to associate a food reward with a visual cue through repeated trials (Extended Data Fig. 1k and Supplementary Video 7). Reversal learning was then measured on the same apparatus by switching the rewarded colour 1 day after the initial learning test.

We compared the birds' performance across all assays at two levels, within and between species. Phylogenetic Bayesian mixed models (MCMCglmm) conducted at the interspecific level using species' mean performance revealed that a species' proficiency on one assay was not

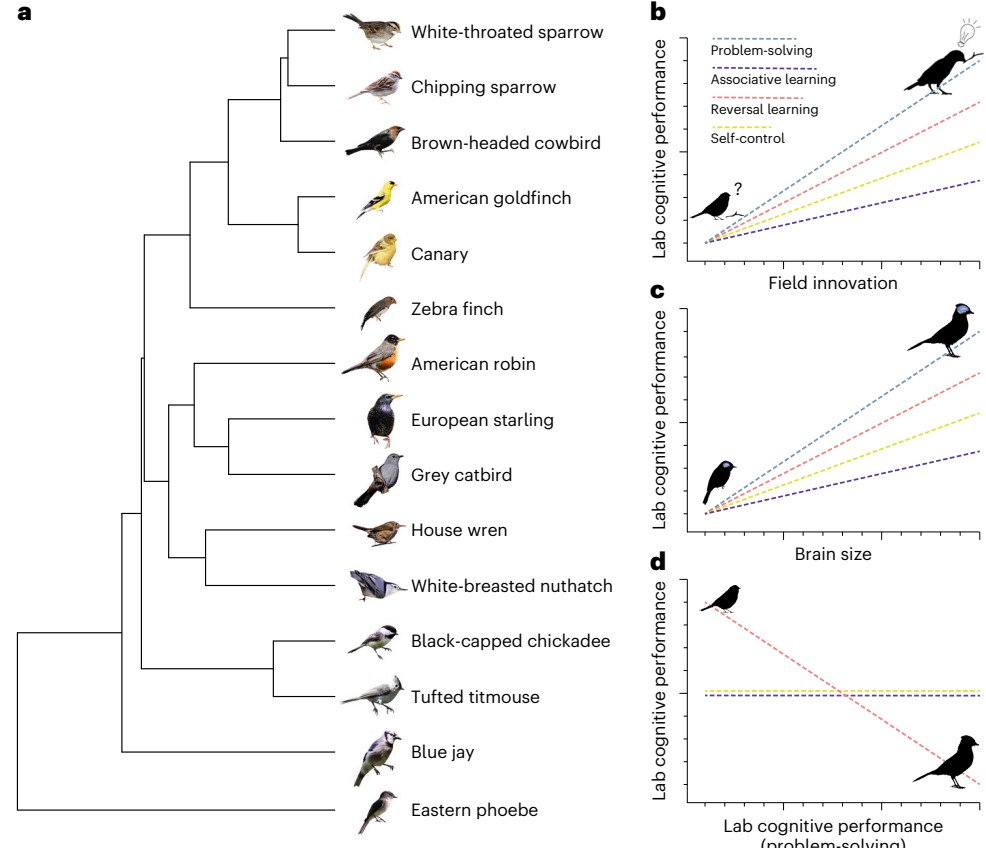

**Fig. 2 | Framework of our study, with the 15 study species and interspecific trait relationship predictions. a**, We assessed cognitive skills in 203 birds from 15 species ($n$ = 12–19 individuals per species). All species belong to the oscine (Passeri) sub-order (songbirds), except the eastern phoebe, a suboscine (Tyranni). Phylogenetic tree relationships were obtained from ref. 53. **b**, Predicted interspecific relationships between field innovation rates and performance on the four cognitive assays tested in this study. According to current assumptions of behavioural flexibility, all cognitive traits should be more or less positively associated with field innovation rates, from non-innovative to highly innovative species; the lowest to highest cognitive performance in our laboratory assays (bottom to top). **c**, Similarly, given the known link between innovation rates and brain size, performance on all assays across species is predicted to be positively associated with brain size. **d**, Problem-solving, which requires persistence, is predicted to be negatively associated with reversal learning, for which persistence reduces performance. No relationships between problem-solving and associative learning or self-control are expected. The predictions in **b** and **c** are likely mutually exclusive of **d**, yet those predictions reflect current knowledge and assumptions. Image credits: Derrick Eidam for wild species and Mélanie Couture for domesticated species (zebra finch and canary) in **a** and Simon Ducatez for the innovative species in **b**; other species' silhouettes are from PhyloPic (http://phylopic.org; chipping sparrow, Ferran Sayol; blue jay, T. Michael Keesey; eastern phoebe, Andy Wilson). Scientific names and sample sizes for each species are provided in Supplementary Table 1.

associated with its performance on the others, except for associative and reversal learning, which showed a significant association (Fig. 3 and Supplementary Table 2b). Species' mean differences in shyness (latency to feed following human disturbance) or neophobia (latency to feed in the presence of a novel object) were not related to performance on any cognitive task (Extended Data Fig. 3 and Supplementary Table 2c).

At the interindividual level, comparisons of each individual's performance on the different cognitive tasks showed no coherent pattern when comparing all 203 individuals using linear mixed models (Supplementary Table 3) or individuals of each of the 15 species separately (Extended Data Figs. 4–8 and Supplementary Table 4). The only exception was the relationship between associative learning and reversal learning tasks that was significantly positive across the 203 individuals (Supplementary Table 3) and within 4 (2 after false discovery rate (FDR) adjustment) of the 15 species (Extended Data Fig. 9 and Supplementary Table 4d), consistent with the interspecific association we found. Thus, testing 203 individuals from 15 species, we find robust support for the idea that the assays, except for the two learning tasks, measure distinct aspects of cognition.

## Problem-solving, innovation rates and brain size

We then asked whether the species' mean performance on the behavioural tests in captivity was predicted by published metrics of field innovations (number of cases of novel observed behaviours, corrected for research effort, that is, the number of articles published per species[3,26]) and brain size (absolute or corrected for body size[27]). We separately analysed food-type innovations (reports of novel food types eaten) and technical innovations (reports of novel foraging techniques[28]). Food-type innovation is suspected to be a consequence of opportunistic generalist foraging[2], whereas technical innovation likely requires problem-solving skills[3]. In line with this prediction, phylogenetic modelling revealed that technical innovation rate in the wild significantly predicted problem-solving performance in our laboratory experiments (Fig. 4a and Supplementary Table 2d). However, we detected no significant association between problem-solving performance and food-type innovation rate (Fig. 4b and Supplementary Table 2d). In addition, problem-solving performance was positively associated with both absolute (Fig. 4c and Supplementary Table 2d) and relative brain size (Fig. 4d and Supplementary Table 2d). The above relationships remained significant when Benjamini–Hochberg FDR corrections

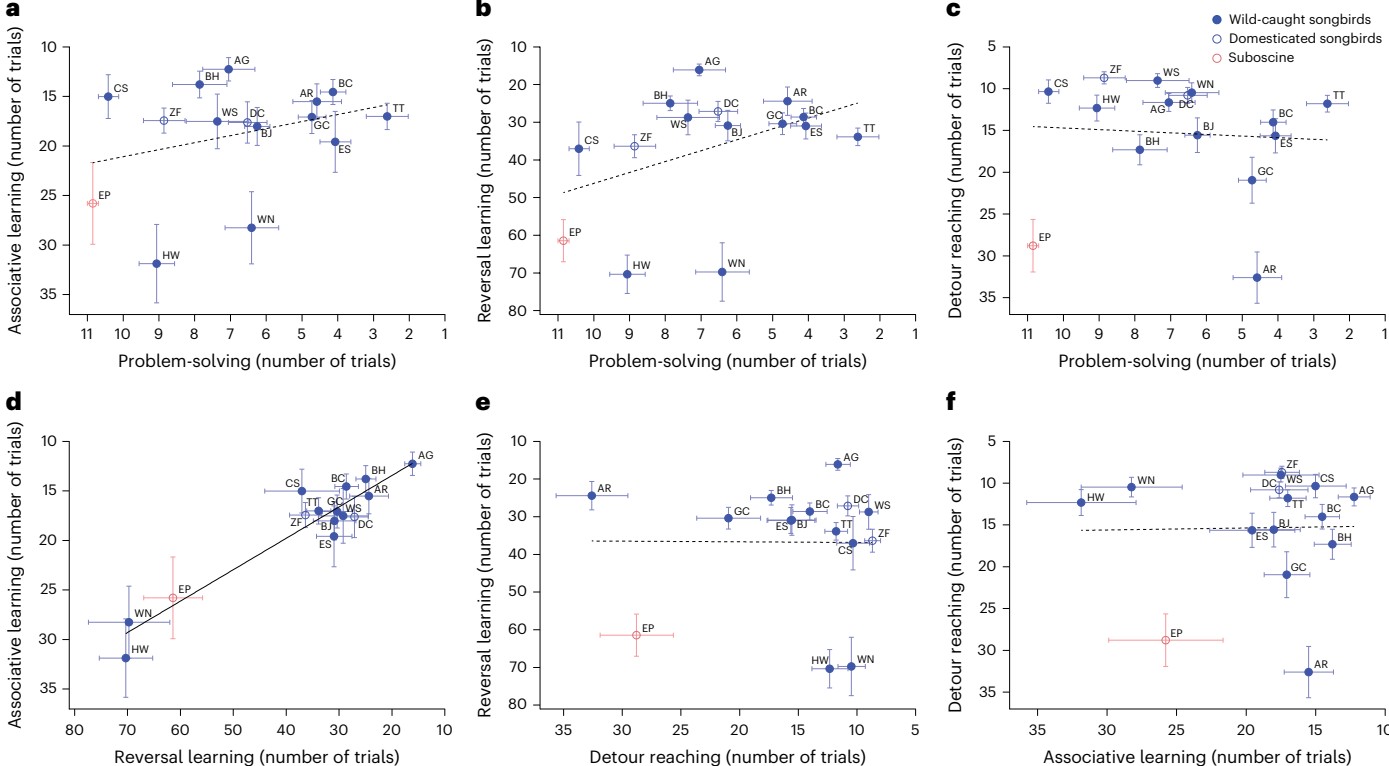

**Fig. 3 | Interspecific relationships between problem-solving, associative learning, reversal learning and self-control. a–c** Problem-solving performance across species is not significantly associated with associative learning (**a**), reversal learning (**b**) or self-control (measured using the detour-reaching task) (**c**). **d**, Associative learning performance is associated with reversal learning performance across species. **e,f**, Self-control performance is not associated with reversal learning (**e**) or associative learning performances (**f**). Problem-solving performance is each species' mean number of trials to solve the four different problems. Graphs illustrate mean species' trial values with s.e.m., ranked predictors and lines of values predicted by Bayesian phylogenetic mixed models. Filled blue circles, wild-caught songbird species; empty blue circles, domesticated songbird species (zebra finch and canary); red circles, the suboscine (eastern phoebe); solid trend line, $P_{MCMC.adj} < 0.05$; dashed trend lines, $P_{MCMC.adj} > 0.05$; species' two-letter codes are listed in Supplementary Table 1; detailed results of MCMCglmm modelling and FDR-corrected $P$ values ($P_{MCMC.adj}$) are provided in Supplementary Table 2b.

were applied (Supplementary Table 2d). In contrast to problem-solving, associative learning, reversal learning and self-control were not associated with innovation rate or brain size (Extended Data Fig. 10 and Supplementary Table 2e–g). These results suggest that the cognitive skills measured by problem-solving tasks in captivity are similar to the ones required to invent technical innovations in the wild and are linked with increased brain size but that these cognitive skills are distinct from those measured by assays of associative learning, reversal learning and self-control.

### Relationships are not driven by non-cognitive factors

We next examined whether non-cognitive variables were responsible for the relationships we found. We used the complete dataset of 203 individuals to implement phylogenetic Bayesian mixed models (MCMCglmm) that included variables of personality traits (shyness and neophobia), experimental testing conditions (wild-caught or domesticated, body condition, bird choice of food reward used in tests, capture site and fasting period), dietary generalism (number of food categories the species consumes in the wild[2]) and phylogeny (Table 1 and Supplementary Table 5).

Full modelling analyses with all the above variables combined in the same model revealed that domesticated species were less shy than wild-caught species (Supplementary Table 5; models 1–4). Individual shyness was negatively associated with problem-solving performance but positively associated with reversal learning and self-control (Supplementary Table 5; models 9–12 and 17–24). In addition, neophobia was negatively associated with associative learning performance (Supplementary

Table 5; models 13–16), fasting time was positively associated with neophobia (Supplementary Table 5; models 5–8) and food reward type was associated with reversal learning (Supplementary Table 5; models 17–20). However, the presence of these covariates in the models did not invalidate the previously found relationships between problem-solving, technical innovation and brain size, all of which remained significant (Supplementary Tables 1 and 5; models 9, 11 and 12). Finally, neophobia was positively associated with food innovation along with captive status, dietary generalism, reward type and fasting time (Supplementary Table 5; model 6), but the relationship with food innovation was not significant following the FDR correction (Table 1). In summary, full modelling analyses revealed that while a few covariates were associated with some cognitive measures, none accounted for the relationships between problem-solving, technical innovation in the wild and brain size.

## Discussion

Our results show that tests of problem-solving in captivity are an appropriate experimental assessment of technical innovativeness in the wild. The other cognitive traits we measured, namely, associative learning, reversal learning and self-control, were unrelated to innovation rates. Traits measured by these assays, as well as innovation in the wild, have all been considered components of behavioural flexibility[16]. Our results show that an umbrella term of this type is not homogeneous. The absence of relationships at the interindividual and interspecific levels suggests that the cognitive traits measured by our tasks are distinct and that only problem-solving assays measure, at least partially, the cognitive skills required to innovate in the wild.

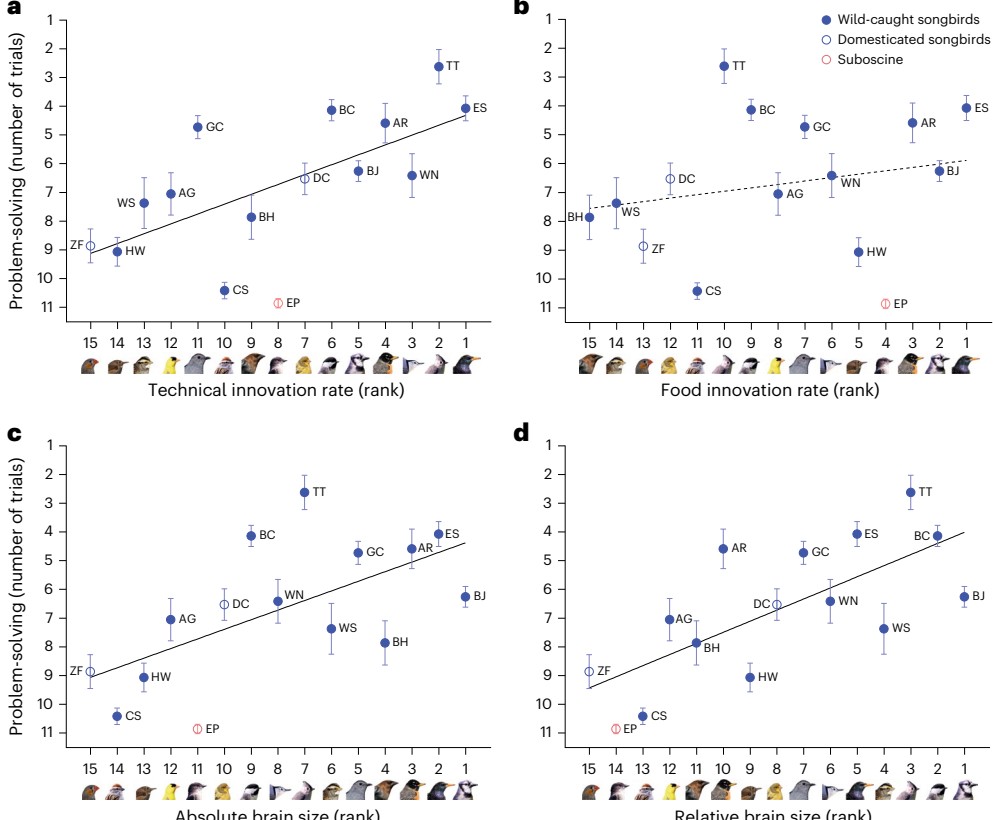

**Fig. 4 | Interspecific relationships between problem-solving, innovation and brain size. a,b**, Mean problem-solving performance across species is significantly associated with their technical innovation rates (**a**) but not with their food innovation rates (**b**). **c,d**, Problem-solving is positively associated with absolute brain size (**c**) and relative brain size across species (**d**). Problem-solving performance is each species' mean number of trials to solve the four different problems; innovation rates are innovation reports corrected for investigator research effort obtained from refs. 3,26; relative brain sizes are the residuals of brain volumes with body weight, and absolute brain sizes are brain volumes; brain size and body weight data were obtained from ref. 27. Graphs illustrate mean species' trial values with s.e.m., ranked predictors and lines of values predicted by Bayesian phylogenetic mixed models. Filled blue circles, wild-caught songbird species; empty blue circles, domesticated songbird species (zebra finch and canary); red circles, the suboscine (eastern phoebe); solid trend lines, $P_{MCMC.adj} < 0.05$; dashed trend line, $P_{MCMC.adj} > 0.05$; species' two-letter codes are listed in Supplementary Table 1; detailed results of MCMCglmm modelling and FDR-corrected $P$ values ($P_{MCMC.adj}$) are provided in Supplementary Table 2d. Image credits: Derrick Eidam for wild species and Mélanie Couture for domesticated species (zebra finch and canary).

Problem-solving is widely recognized as a hallmark of human executive functions[29]. Still, its assessment in animals has been the object of a number of questions concerning its cognitive nature, as well as its biological and ecological relevance (for example, ref. 30). Our results suggest that potential confounding variables such as shyness, neophobia or experimental conditions are not responsible per se for the interspecific variation observed, consistent with previous evidence at the intraspecific level (for example, ref. 10). Instead, technical innovation rate in the wild and brain size, both considered ecologically relevant metrics, are highly predictive of problem-solving performance. In fact, innovation rate is linked with a lower risk of extinction[3] and a greater colonization success[1]. Absolute and relative brain sizes are the only measures of a neural substrate available for our 15 species, but they are closely linked to finer measures such as neuron numbers[7,31], the volume of the associative pallium[32] and expression levels of neurotransmitter receptors, which are all associated with innovation[7,33,34].

Our study was conducted with the largest sample of avian species, individuals and assays thus far. Two notable studies on mammals have examined similar questions on large taxonomic samples. Performance on obstacle removal problems has been assessed in 39 species of captive carnivores and, as we found here, is associated with brain size[35]. Self-control performance was compared on a taxonomically heterogeneous sample of 36 species ranging from elephants to zebra finches[13];

a subset of this analysis focused on a more homogeneous sample of 23 primate species and showed positive associations between self-control, brain size and innovation rate taken from a published database[5]. Given that the two mammal studies used only one type of assay, it is difficult to judge whether birds and mammals differ in the way brains, innovations and experimental assays are connected. Considering the remarkable degree of convergent evolution between birds and primates[36,37], comparing primates on different assays is an obvious next step, given the known relationship between various experimental tasks[38,39] and field-based counts of cognition[6].

In a review, Griffin and Guez[24] concluded that extractive foraging problems were a good experimental measure of innovativeness in the wild and that the diversity of the motor acts used in solving a problem was a critical factor in success[40,41]. In line with this hypothesis, diversity of technical innovations is a better predictor of relative brain size than any other measure on a broad sample of avian species from 76 families[28]. In our study, the fact that technical innovation rate in the wild is the only significant predictor of problem-solving suggests that trying out a diversity of motor solutions to a foraging problem in both captivity and the wild is more important than inhibiting an initial unproductive response (measured by self-control assays) or learning about cue changes (measured by associative and reversal learning assays). This is also the route, inspired by animal studies, taken by recent work

**Table 1 | Summary of MCMCglmm final models assessing relationships between all measured cognitive traits and innovation or brain size**

| Model | Dependent variable | Independent variable | post.mean | $_{l-95\%}$ C.I. | $_{u-95\%}$ C.I. | $P_{MCMC}$ | $P_{MCMC.adj}$ |
|---|---|---|---|---|---|---|---|
| 1 | Shyness | Technical innovation | −0.131 | −0.322 | 0.059 | 0.166 | 0.663 |
| 2 | Shyness | Food innovation | 0.009 | −0.180 | 0.202 | 0.931 | 0.931 |
| 3 | Shyness | Absolute brain size | 0.070 | −0.535 | 0.676 | 0.807 | 0.931 |
| 4 | Shyness | Relative brain size | 0.315 | −0.376 | 1.020 | 0.352 | 0.703 |
| 5 | Neophobia | Technical innovation | −0.041 | −0.325 | 0.244 | 0.754 | 0.919 |
| 6 | Neophobia | Food innovation | −0.267 | −0.508 | −0.029 | 0.029 | 0.116 |
| 7 | Neophobia | Absolute brain size | −0.222 | −1.080 | 0.681 | 0.583 | 0.919 |
| 8 | Neophobia | Relative brain size | −0.026 | −1.003 | 1.005 | 0.919 | 0.919 |
| 9 | Problem-solving | Technical innovation | **−0.204** | **−0.362** | **−0.047** | **0.016** | **0.021** |
| 10 | Problem-solving | Food innovation | −0.046 | −0.224 | 0.132 | 0.584 | 0.584 |
| 11 | Problem-solving | Absolute brain size | **−0.587** | **−1.015** | **−0.165** | **0.011** | **0.021** |
| 12 | Problem-solving | Relative brain size | **−0.671** | **−1.188** | **−0.153** | **0.016** | **0.021** |
| 13 | Associative learning | Technical innovation | −0.014 | −0.107 | 0.080 | 0.758 | 0.758 |
| 14 | Associative learning | Food innovation | 0.034 | −0.051 | 0.118 | 0.403 | 0.758 |
| 15 | Associative learning | Absolute brain size | −0.109 | −0.379 | 0.158 | 0.405 | 0.758 |
| 16 | Associative learning | Relative brain size | −0.050 | −0.371 | 0.272 | 0.744 | 0.758 |
| 17 | Reversal learning | Technical innovation | −0.050 | −0.180 | 0.079 | 0.422 | 0.562 |
| 18 | Reversal learning | Food innovation | 0.003 | −0.120 | 0.125 | 0.954 | 0.954 |
| 19 | Reversal learning | Absolute brain size | −0.277 | −0.619 | 0.067 | 0.110 | 0.440 |
| 20 | Reversal learning | Relative brain size | −0.224 | −0.658 | 0.205 | 0.287 | 0.562 |
| 21 | Self-control | Technical innovation | 0.034 | −0.108 | 0.177 | 0.622 | 0.830 |
| 22 | Self-control | Food innovation | 0.059 | −0.071 | 0.188 | 0.348 | 0.697 |
| 23 | Self-control | Absolute brain size | 0.273 | −0.101 | 0.658 | 0.147 | 0.590 |
| 24 | Self-control | Relative brain size | −0.010 | −0.491 | 0.477 | 0.965 | 0.965 |

The effects of published metrics were tested along with significant covariates, if any (not shown here; see Supplementary Table 5 for details). All measured cognitive traits are expressed in trial numbers to succeed (logged); therefore, negative effects with innovation or brain size metrics indicate positive relationships (for example, higher problem-solving performance is associated with higher innovation rates). Innovation variables are corrected for species' research effort. post.mean, posterior mean; $_{l-95\%}$C.I., lower 95% confidence interval; $_{u-95\%}$C.I., upper 95% confidence interval; $P_{MCMC}$, MCMC $P$ value; $P_{MCMC.adj}$, FDR-corrected MCMC $P$ value. Bold values denote significant relationships after FDR corrections. Sample size=203 individuals, 15 species.

in engineering, where motor diversity helps robots creatively solve problems for which they were not initially programmed[42].

Taken together, our results validate long-standing but untested hypotheses concerning the links between problem-solving, innovation and the brain but question the assumption that behavioural flexibility can be concomitantly operationalized through innovation reports and assays of problem-solving, associative learning, reversal learning and self-control. The obvious next step is to examine in more detail which brain components are responsible for increased problem-solving skills in some species, which allow them to be more successful in changing environments.

## Methods

### Wild bird captures and acquisition of domestic birds

We caught 178 wild birds of 13 species between 2018 and 2020 (March to December) at the Rockefeller University Field Research Center (RUFRC) in Millbrook, NY, USA (latitude, 41° 46′ 3.0″ N; longitude, 73° 45′ 2.5″ W). Permits were obtained from all university and government instances for bird captures and experiments. Birds were captured using mist nets placed in 8 sites (4 open and 4 forested, each at 200 to 500 m) within a 30 ha area around the RUFRC main campus. The captured birds were weighed, measured, banded, sex-typed, and then placed into their behavioural cages.

In addition to the wild birds, we included 25 birds of 2 domesticated species. Twelve zebra finches aged between 9 and 15 months were

obtained from the domestic colony at RUFRC, and 13 'American Singer' canaries aged between 8 and 16 months were purchased from Stewart's bird farms. Domesticated birds underwent the same processes as wild birds upon arrival at the behaviour laboratory.

Initially, only male birds were selected to enhance statistical power by minimizing potential variations in behaviour driven by sex. However, because capturing sufficient numbers of blue jays and European starlings proved challenging in our capture area, we included female birds of these species in the study ($n$ = 3 for blue jays and $n$ = 4 for European starlings) as no significant cognitive differences were found compared to male birds of these species (Supplementary Table 6).

### Morphometric measurements

Standard measurements were taken using the identical procedure described in ref. 10. These measurements were taken by a single individual (J.-N.A.). To assess body condition, scaled mass index was calculated for each individual using wing length and body weight, following the procedure described in ref. 43.

### Housing conditions

Birds were housed individually in custom-designed aluminium cages measuring 81.3 cm × 55.9 cm × 68.6 cm in an indoor aviary at the RUFRC. Birds were visually (but not acoustically) isolated from each other by opaque plastic panels. A Brio 4K Ultra-HD camera (Logitech) was used to video-record and live-view all observations in an adjacent room

where the birds could not see or hear the experimenter, which remained the same (J.-N.A.) throughout all behavioural tests.

To maintain the birds' circadian rhythm and minimize stress, the daily lighting conditions in the aviary were adjusted to follow the natural light cycle. During the first three captivity days (Friday to Sunday), the birds were undisturbed, except for the daily replenishment of food and water. They had unrestricted access to water and food, which included sunflower seeds (Ultra Clean sunflowers, Kent Nutrition Group), mealworms (Bug Company), wax worms (Bug Company) and species-specific seed mix (blue jays and European starlings, wild bird mix of seeds, grain and nuts; canaries, canary seed mix; American goldfinches, half-and-half mix of thistle and canary seed mix; other species, finch seed mix (Blue Seal Neat Feast, Blue Seal Colours 'n Chorus canary diet, Blue Seal Colours 'n Chorus Finch Diet, Kent Nutrition Group)).

If any bird did not eat or displayed distress signs, it was immediately released. At the end of the captivity period, all birds, except for a few individuals killed for tissue sampling for another study, were released back to their initial capture site.

### Molecular sexing

We determined the sex of all individuals, using a standard sex-typing PCR protocol[44,45]. Briefly, we collected approximately 20 µl of blood by puncturing the brachial vein. A PCR was run using 1 µl of DNA extracted from blood samples, and the amplified DNA was migrated on a 2% agarose gel.

### Behavioural tests

**General procedure.** Following the 3 day habituation period, the birds underwent our 6 day behavioural testing procedure in the same cage. They were food-deprived overnight before each testing day to ensure sufficient participation in the behavioural tests. We adjusted the deprivation period according to each bird's body weight and night lengths throughout the seasons, both being expected to impact the fasting state. We used the following formula that we developed[22]: Deprivation time (h) = 2 × ln (Bodyweight (g)) + 0.2 × Night length (h) + 7. The same formula was applied each day to calculate the fasting period for each individual.

All feeding dishes and behavioural tasks were constructed in three different sizes and mounted on standardized white acrylic base plates: small (100 mm × 100 mm), medium (125 mm × 125 mm) and large (165 mm × 165 mm). Small apparatuses were used for birds weighing 10 to 20 g; medium, 21 to 40 g; and large, 41 to 85 g. On the first testing day, after food deprivation but before the behavioural tests, the birds were presented with three types of food (seed mix, mealworms and softened dog food pellets) to determine their preferred food, utilized as their reward in all behavioural tests. No bird chose dog food, 143 birds chose mealworms and 60 chose seeds.

The order of the behavioural tests was fixed for all birds to minimize the influence of test order on bird performance[46]. Only one cognitive test was conducted each day, except for the last day (see Cognitive tests). The first four days also included personality measurements before the cognitive tests. Problem-solving tasks considered 'easier' were presented at the beginning (days 1 and 2), while more challenging tasks were presented at the end of the captivity period (day 6) to increase the overall probability of success. The tests started between 7:00 and 11:00, depending on the calculated fasting period for each bird, and concluded no later than 16:00. Then, the birds were allowed to feed ad libitum until the start of the subsequent overnight deprivation. A 5 min pause was given between each trial for all tests.

**Personality measurements.** Shyness was the first measurement taken on the first four testing days. The feeding dish (Extended Data Fig. 1a) was introduced into the cage, and the experimenter immediately left and started a stopwatch. The latency (in seconds) to feed was recorded when the birds first contacted the food. No maximum latency cap was

set for shyness trials. The birds were allowed to feed for 15 s before the food was removed from the cage. Shyness was measured again after the neophobia assessment (see below). The average of the two shyness measurements was calculated for each day, and the shyness variable used in the analyses was the mean of these four shyness values. While shyness decreased from day 1 to day 4 due to habituation to the experimenter (means ± s.e.m.; day 1, 415.96 ± 65.80 s; day 2, 88.51 ± 33.41 s; day 3, 38.78 ± 8.90 s; day 4, 23.74 ± 5.23 s; $n$ = 203), the effect was consistent across all species (slope across 4 days for all species, mean ± s.e.m., −1.13 ± 0.12). Taking into account the effect of the test day and species, shyness measurements were repeatable[47] across the four days for each individual (Supplementary Table 7).

After a 5 min pause, neophobia was assessed by presenting a novel object beside the feeding dish and recording the latency to feed. The mean shyness latency for that day was subtracted from the neophobia latency to obtain a measure of 'pure' neophobia. This procedure was repeated for four consecutive days, with a different novel object introduced each day (day 1, four coloured cotton balls, Extended Data Fig. 1b; day 2, two stacks of coloured Duplo blocks, Extended Data Fig. 1c; day 3, two Erlenmeyer flasks with coloured tapes, Extended Data Fig. 1d; day 4, one inflated purple glove, Extended Data Fig. 1e). We used three sets (small/medium/large) of neophobia objects matched to the body mass category of the species (see General procedure). The maximum allotted latency to feed was 2 h; if the birds did not feed before this limit, their recorded latency was 7,201 s (which occurred 22 times out of the 812 neophobia trials). Neophobia measurements were repeatable across the four days (Supplementary Table 7). We used the average of the four neophobia measurements in statistical analyses.

**Cognitive tests.** To measure problem-solving, we presented the birds with novel problems that they had to solve on their own, without any previous training or shaping. This method differs from some studies that utilize training procedures on 'novel foraging tasks' (also known as 'shaping' or 'stage-learning'; for example, ref. 48) and then assess how well the animals are capable of repeating the solution. Applying a previously learned solution likely involves different cognitive processes than solving a novel problem. To enhance the precision of our problem-solving measurement, we implemented four different problems, each built in three sizes to match the body mass category of each species.

On the first day, we presented the 'lid-pulling' problem-solving test, consisting of a glass flask containing a food reward, sealed with a loose cork lid that could be removed by pecking its sides or grabbing the top wooden tip (Extended Data Fig. 1f and Supplementary Video 1). To reduce neophobia toward the task, the apparatus was first presented open and left inside the cage until the birds consumed the reward. After a 5 min pause, the task was presented closed and left in the cage for 5 min or until the bird solved the problem. If unsuccessful, the birds were given a 5 min pause, after which the following trial commenced. Birds that failed to solve the task within 10 trials were considered unsuccessful and assigned an arbitrary value of 11 trials. Birds that succeeded underwent the task again to confirm their success. Of the successful birds, 91.3% (84/92) solved the problem a second time. The same testing procedure was used for the following problem-solving tasks.

On the second day, we presented the 'lid-flipping' problem, which consisted of a transparent plastic container loosely closed with a flat plastic lid (Extended Data Fig. 1g and Supplementary Video 2). The birds could solve this problem by grabbing the lid from the side or pecking it from bottom to top. Out of the birds that succeeded in solving the lid-flipping problem within 10 trials, 99.1% (107/108) solved it again upon the second presentation of the task.

On the third day, we evaluated self-control using a detour-reaching task (Extended Data Fig. 1j and Supplementary Videos 5 and 6), following a standard procedure[34]. The birds first underwent a training phase, during which they only had to consume a reward inside an opaque

cylinder, without any time limit. After seven trials, they advanced to the testing phase, which used an identical but transparent cylinder. In this phase, the birds had to reach directly for the reward without pecking at any part of the cylinder to succeed. The success criterion was seven consecutive successful trials, and the maximum allotted trial number was 50; after that, the birds were given a score of 51 trials if unsuccessful.

On the fourth day, we presented a colour-discrimination associative learning task using an apparatus identical to the lid-flipping task but painted entirely yellow or green (Extended Data Fig. 1k and Supplementary Video 7). Before proceeding with the associative learning test, we ensured that all birds were capable of removing the lids from the containers. Birds that did not solve the task during the problem-solving procedure on day 2 were trained until they mastered it using the shaping procedure described in ref. 34, without a maximum trial limit. In brief, the task was presented in progressively harder steps: open, half-closed, three-quarter closed, closed upside down and finally fully closed. Each step had to be completed twice before progressing to the next step. Finally, all birds were given five additional practice trials. At the end of this training phase, all birds could flip lids efficiently.

The associative learning procedure was similar to ref. 46. To familiarize the birds with the task, two open lid-flipping apparatuses (one green, one yellow) were placed on each lateral end of the cage and left inside until the birds fed from both. After a 5 min pause, the apparatuses were presented in switched positions but with closed lids. They were left in the cage until the birds opened and fed from both. Next, the birds underwent a colour choice trial to account for potential colour preferences. The tasks were presented closed and were removed after the birds ate from the first opened apparatus, which was considered their preferred colour. The reward was placed in the non-preferred colour for the subsequent trials. The apparatuses were then presented closed in alternating positions for each trial and were removed immediately if the birds chose the non-rewarded colour or after allowing them to eat the reward (worm or seeds) for 10 s if they chose the rewarded colour. The success criterion for associative learning was 7 consecutive correct trials, excluding the training trials; thus, the best possible score was 7 trials. This task had no maximum trial number to ensure that all birds learned the initial colour before proceeding to the subsequent reversal learning test.

On the fifth day, a reversal learning test was conducted using the same associative learning apparatus and procedure. However, the colours were switched: the previously rewarded colour was now non-rewarded, and vice versa. The success criterion was seven consecutive correct trials. Birds that failed to meet this criterion within 100 trials were given a score of 101 trials.

On the sixth day, we presented two additional problem-solving tasks. First, the 'lid-piercing' problem consisted of a transparent plastic container covered with a piece of aluminium foil secured with a rubber band (Extended Data Fig. 1h and Supplementary Video 3). The birds had to pierce or tear the aluminium foil to access the reward. After the second presentation, all successful birds (139 out of 139) solved the lid-piercing problem again.

On the same day, we presented the 'stick-pulling' problem-solving task, a transparent plastic container attached to a wooden stick and inserted into a transparent plexiglass enclosure (Extended Data Fig. 1l and Supplementary Video 4). The birds had to pull the stick to access the container and remove the lid to obtain the reward. Among the successful birds, 88.8% (71/80) solved the stick-pulling problem a second time.

### Innovation and brain size data

Innovation values were obtained from the most recent innovation database[3,49]. Innovations are published cases of novel feeding (incorporation of an unusual or previously unknown food source in the animal's diet) or technical (use of a novel foraging technique) behaviours in the literature, based on the presence in the report of keywords such as 'new',

'never observed', 'first report', 'opportunistic' and so on. A standard practice when using innovation databases is to correct innovation rates with research effort, that is, the number of articles published for each species[3], as the probability of observing an innovation increases with the time spent observing a species[28]. We used the residuals of a linear model with logged numbers of innovations (food type or technical) as the dependent variable and logged research effort as a fixed independent effect.

Species data for brain size and body mass (average for both sexes when available) were collected from ref. 27. Relative brain sizes were calculated using the residuals from a linear model with logged brain volumes as the dependent variable and logged body mass as the fixed independent effect. Brain size data for the chipping sparrow (*Spizella passerina*) was unavailable. Therefore, we used the brain volume of its closest relative, the American tree sparrow (*Spizella arborea*) and scaled it proportionally with the body size difference between the two species. Excluding the chipping sparrow did not change the outcomes of our brain size analyses. Brain size data for individual sexes are not available.

### Statistical analyses

All statistical analyses were conducted in R version 4.3.0[50]. Trial numbers to success criterion were used for all cognitive tasks. Using latencies instead yielded similar results. The average trial number to solve the four problems was used in our models as they were all strongly associated (Extended Data Fig. 2 and Supplementary Table 2a).

The repeatability of the shyness and neophobia measurements were calculated with the RptR 0.9.22 package[51] using individual measurements as the dependent variable, the measurement day (1–4) and species as fixed effects, and the bird identity as the grouping variable and random effect ($n$ = 203 individuals; each personality trait was measured 4 times).

All interspecific relationships were assessed with phylogenetic Bayesian models using the MCMCglmm[52] package in R. We conducted models with each species' mean cognitive performance (trials) set as the dependent and independent variables (with no other covariables, as opposed to the full modelling strategy; see below), with phylogenetic distance and captive status (wild-caught or domesticated) set as random effects. The MCMCglmm parameters can be found in the available code. The models were repeated 100 times, and the values from all runs were averaged. A single consensus phylogenetic tree, obtained from ref. 53, was used for phylogeny calculations in the models.

Interindividual relationships were assessed with linear mixed models (lmer) in R using the complete dataset of 203 values, with each trait set as either dependent or independent variables, and species as a random variable. The results were then validated using corresponding MCMCglmm with 'species' added as a random effect, in addition to the random 'animal' phylogenetic term to account for phylogeny. We also assessed interindividual relationships between cognitive traits within each species by running simple linear models (lm) in R for each species separately.

We then explored relationships between each trait of interest by performing full models using MCMCglmm, this time with the whole dataset of 203 individual logged values instead of species means, and with potential covariates included. Phylogenetic relationships (to account for non-equivalent phylogenetic distance among all species), species identity (to account for repeated testing of each species) and capture sites (eight levels for capture locations, to account for potential relatedness of the individuals or any other ecological factor linking individuals) were included as random effects. Fasting time, reward type, body condition, shyness, neophobia, captive status and dietary generalism[2] were included in all models as fixed effects. Because higher trial numbers represent lower performance in cognitive tasks, results of negative estimates with published metrics (innovation rates and brain size) represent positive relationships (for example, more

innovative species solve problems in fewer trials). For all models, we verified that autocorrelation was below an acceptable threshold (all were <0.1) using the 'autocorr.diag' function and by visualizing the plots of the posterior distributions of the variance components of our models. Each full model was run 100 times, and we report the means of all values in Supplementary Table 5 ('a' models). We then removed non-significant fixed effects to increase the fit of the models. We first removed variables with the highest $P$ values in the initial models, the models were rerun and the process was repeated until only significant variables ($P < 0.05$) remained in the final models. When only significant variables remained, we reran the models 100 times to report the final model mean values (Supplementary Table 5, 'b' models). Table 1 reports model results for only variables of interest (metrics of innovation and brain size) obtained in final models (run with significant covariables, if any). When no variable of interest remained (thus being absent in Supplementary Table 5, 'b' models), we reran the models 100 times with innovation and brain size variables re-added to obtain their estimates.

$P$ values from all analyses (except the full MCMCglmm models of Supplementary Table 5) were adjusted to account for multiple testing with the Benjamini–Hochberg[54] FDR correction using the 'p.adjust' function in R, 'BH' method. $P$ values were grouped by blocks of similar analysed data to perform the adjustments (each panel of Extended Tables 2–4 constituted a separate block; for example, Supplementary Table 2a constituted a block of 6 corrected $P$ values; also see available code). The significance threshold was set at $P = 0.05$.

## Notes on study species selection

This study aimed to compare cognitive traits in songbird species, which show relatively homogenous morphologies. Our behavioural tasks required the birds to perform motor actions; therefore, including birds from more phylogenetically distant clades would likely increase morphological variation (for example, species with curved/thin beaks, species that rely on their legs to manipulate objects and so on), which would have complicated the interpretation of our results as the outcomes could have been influenced by morphology rather than cognition. We also added a closely related non-songbird species (eastern phoebe, a suboscine) as it was abundant in our capture area and its morphology was sufficiently similar to our other study species to allow it to perform in our behavioural tasks. However, we were cautious when interpreting data on this species; separate tests that excluded the suboscine did not change the outcome of our analyses. In addition, all analyses controlled for phylogenetic distance. The 13 wild-caught study species were chosen from a total of 21 species caught and tested during the first season, based on their feasibility of capture (sufficient number of caught birds per species to achieve a minimum of $n = 12$ male birds), assessed at the end of the first year of capture. We did not include data from the species for which only one or two birds per species were tested because our study aimed to provide as robust a test as possible of the different behavioural assays, based on a large sample of subjects per species. In another study conducted in parallel focusing on the link between problem-solving and vocal learning complexity[22], we included problem-solving data from all 21 wild-caught species to verify whether our conclusions held when looking at more species, but they were not included in the present study for the above reason. In addition to the 13 wild bird species, we tested two domesticated species, the canary and the zebra finch. These two species are by far the most studied songbirds. We believe that including well-characterized birds raised in the same conditions provided an opportunity to generate valuable knowledge. Still, we were also cautious when interpreting data from those species, as domestication could have affected relationships between traits. Performing all analyses without these two species did not change the outcomes. We also included a 'captive status' variable (wild-caught or domesticated) in our models to account for those potential differences.

## Reporting summary

Further information on research design is available in the Nature Portfolio Reporting Summary linked to this article.

## Data availability

The raw dataset is available at https://zenodo.org/records/10206756.

## Code availability

The code scripts are available at https://zenodo.org/records/10206756.

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

## Acknowledgements

We thank S. Ducatez and F. Sayol for help with the statistical analyses and L. Cauchard for insightful discussions on the manuscript. This study was supported by the Howard Hughes Medical Institute and the Rockefeller University, a Banting postdoctoral fellowship and an Natural Sciences and Engineering Research Council postdoctoral fellowship to J.-N.A., and Howard Hughes Medical Institute funds and a National Institutes of Health Director's Transformative Research Award to E.D.J.

## Author contributions

Conceptualization: J.-N.A., M.C., L.L., E.D.J. Methodology: J.-N.A., M.C., L.L. Investigation: J.-N.A., M.C. Visualization: J.-N.A., M.C. Funding acquisition: J.-N.A., E.D.J. Supervision: J.-N.A., E.D.J. Writing—original draft: J.-N.A., L.L. Writing—review and editing: J.-N.A., M.C., L.L., E.D.J.

## Competing interests

The authors declare no competing interests.

## Additional information

**Extended data** is available for this paper at https://doi.org/10.1038/s41559-024-02342-7.

**Correspondence and requests for materials** should be addressed to Jean-Nicolas Audet.

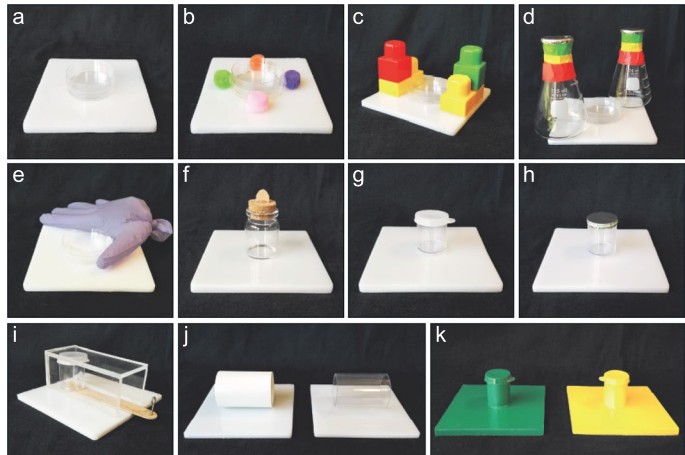

**Extended Data Fig. 1 | Behavioral tasks used to assess behaviour in the 15 species.** (**a**) Feeding dish used throughout the captivity period, including for shyness assessment. (**b-e**) Novel objects used to assess neophobia on days 1 to 4, respectively. (**F**) 'Lid-pulling' problem-solving task. (**g**) 'Lid-flipping' problem-solving task. (**h**) 'Lid-piercing' problem-solving task. (**i**) 'Stick-pulling' problem-solving task. (**j**) Detour reaching task. An opaque cylinder was used for the training phase (left), and a transparent cylinder for the testing phase (right). (**k**) Color discrimination learning apparatus used to assess associative and reversal learning. All tasks have been constructed in three sizes, matching the body size of the tested birds. Image credits: Mélanie Couture and Jean-Nicolas Audet for all pictures.

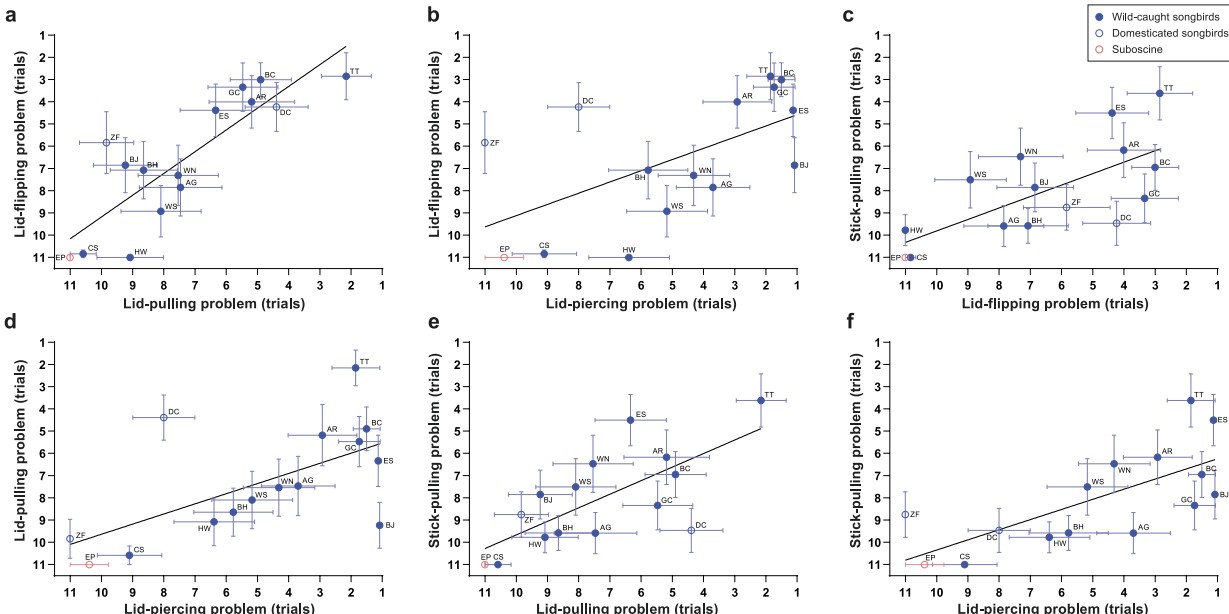

**Extended Data Fig. 2 | Comparisons of the 15 species' performance on each problem-solving task.** Species that solve the lid-flipping problem in fewer trials are also quicker to solve (**a**) lid-pulling, (**b**) lid-piercing, and (**c**) stick-pulling problems. Performance on the lid-pulling problem is also associated with (**d**) lid-piercing and (**e**) stick-pulling problems' performance, and (**f**) stick-pulling with lid-piercing problems' performance. Graphs illustrate mean species' trial values with SEM and lines of values predicted by Bayesian phylogenetic mixed models. Filled blue circles, wild-caught songbird species; empty blue circles, domesticated songbird species (zebra finch and canary); red circles, the suboscine (Eastern phoebe); solid trend lines: $P_{MCMC.adj} < 0.05$; species' two-letter codes are listed in Supplementary Table 1; detailed results of MCMCglmm modelling and FDR-corrected p-values ($P_{MCMC.adj}$) are provided in Supplementary Table 2a.

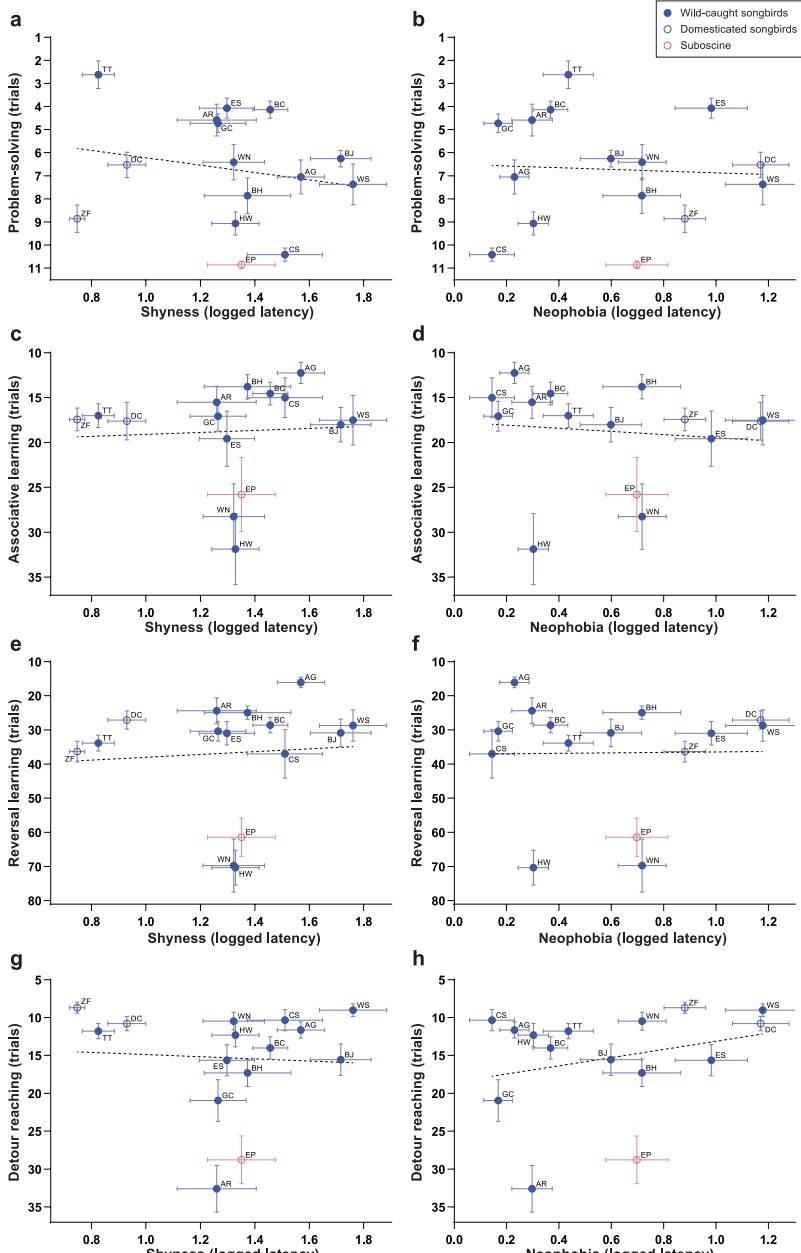

**Extended Data Fig. 3 | Interspecific relationships between cognitive and personality traits.** Species' average problem-solving performance is not associated with their average (**a**) shyness or (**b**) neophobia. Species' average associative learning performance is not associated with their (**c**) shyness or (**d**) neophobia. Species' average reversal learning performance is not associated with their (**e**) shyness or (**f**) neophobia. Self-control performance (detour-reaching task) is not associated with their (**g**) shyness or (**h**) neophobia. Graphs illustrate mean species' trial values, logged personality latencies, and lines of values predicted by Bayesian phylogenetic mixed models. Filled blue circles, wild-caught songbird species; empty blue circles, domesticated songbird species (zebra finch and canary); red circles, the suboscine (Eastern phoebe); error bars: SEM; dashed trend lines: $P_{MCMC.adj}$ > 0.05; species' two-letter codes are listed in Supplementary Table 1; detailed results of MCMCglmm modelling and FDR-corrected p-values ($P_{MCMC.adj}$) are provided in Supplementary Table 2c.

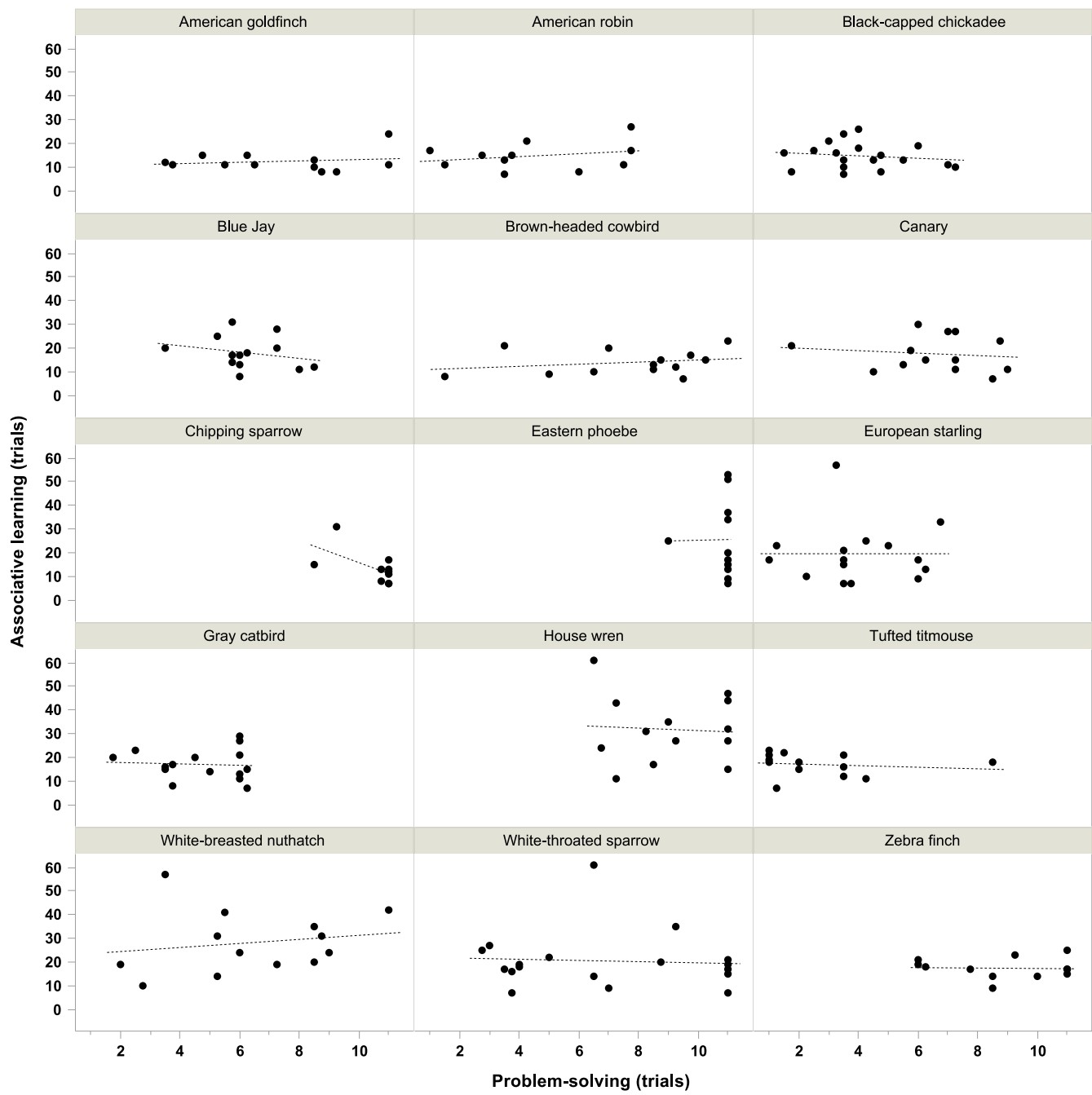

**Extended Data Fig. 4 | Intraspecific relationships between associative learning and problem-solving performance for each of the 15 species.** None of the species shows a significant association between associative learning and problem-solving (all *P* > 0.05). P-values were obtained from linear models (details in Supplementary Table 4a); dashed regression lines: *P* > 0.05.

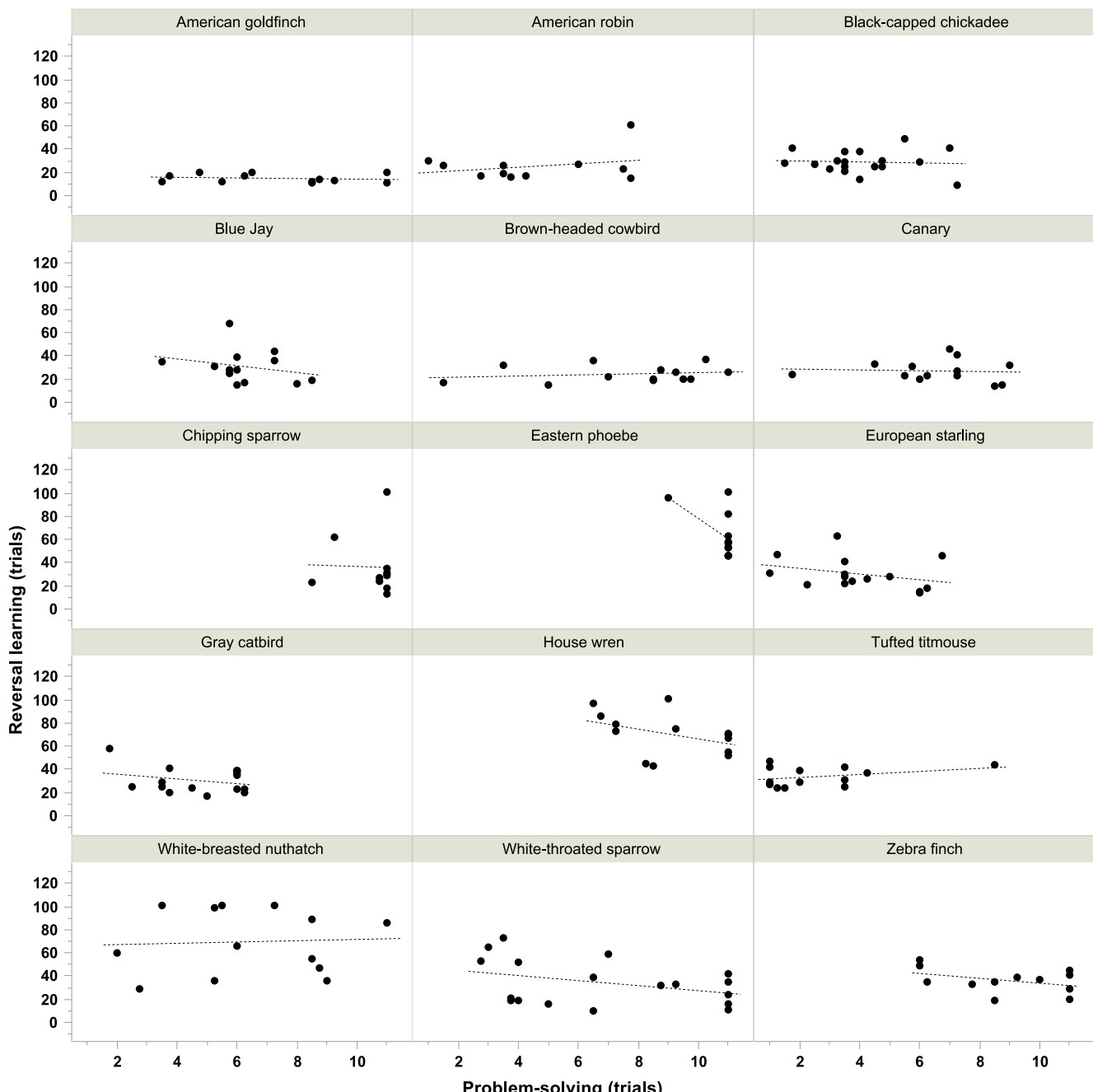

**Extended Data Fig. 5 | Intraspecific relationships between reversal learning and problem-solving performance for each of the 15 species.** None of the species shows a significant association between reversal learning and problem-solving (all *P* > 0.05). P-values were obtained from linear models (details in Supplementary Table 4b); dashed regression lines: *P* > 0.05.

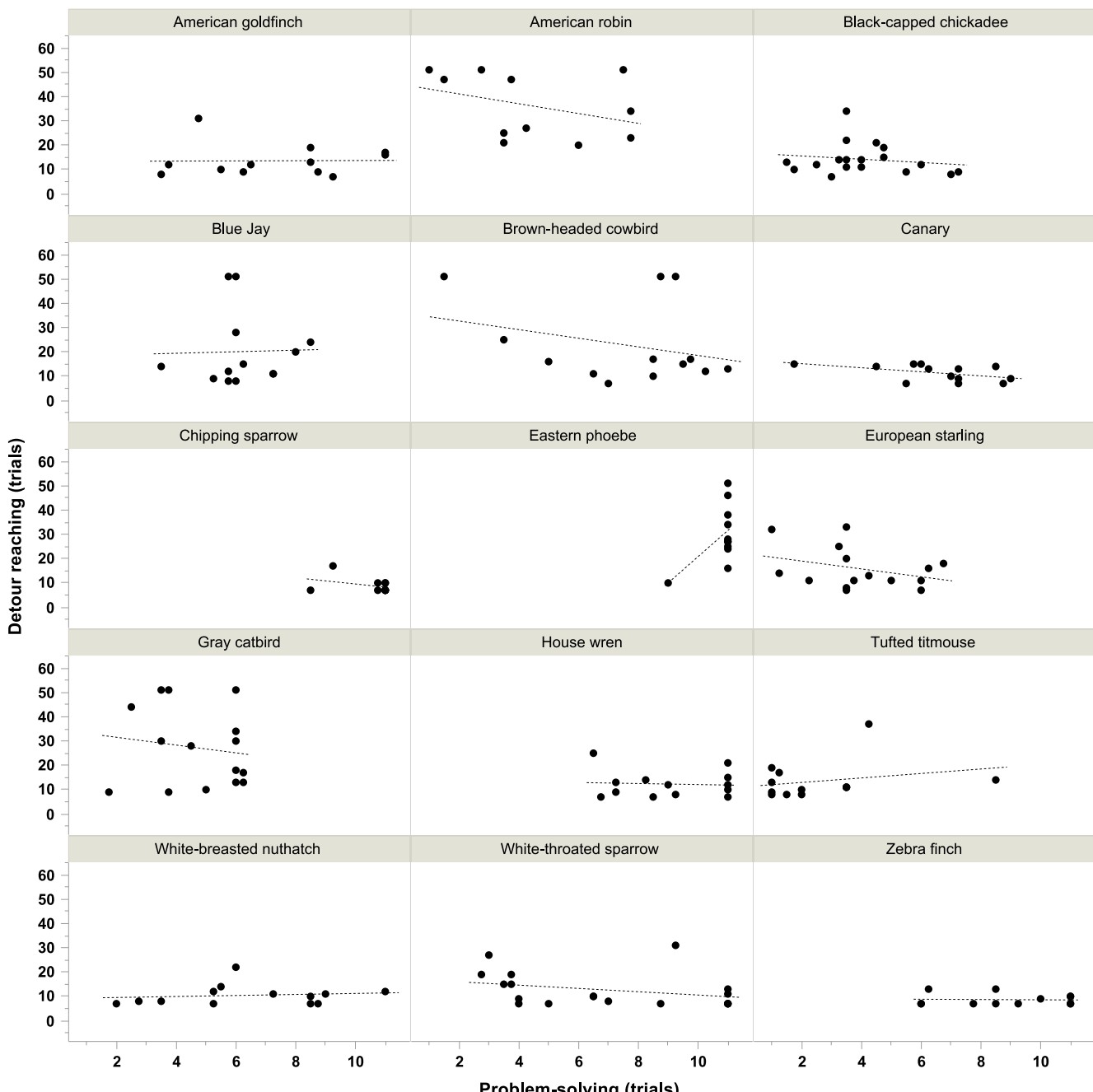

**Extended Data Fig. 6 | Intraspecific relationships between detour reaching and problem-solving performance for each of the 15 species.** None of the species shows a significant association between detour reaching and problem-solving (all *P* > 0.05). P-values were obtained from linear models (details in Supplementary Table 4c); dashed regression lines: *P* > 0.05.

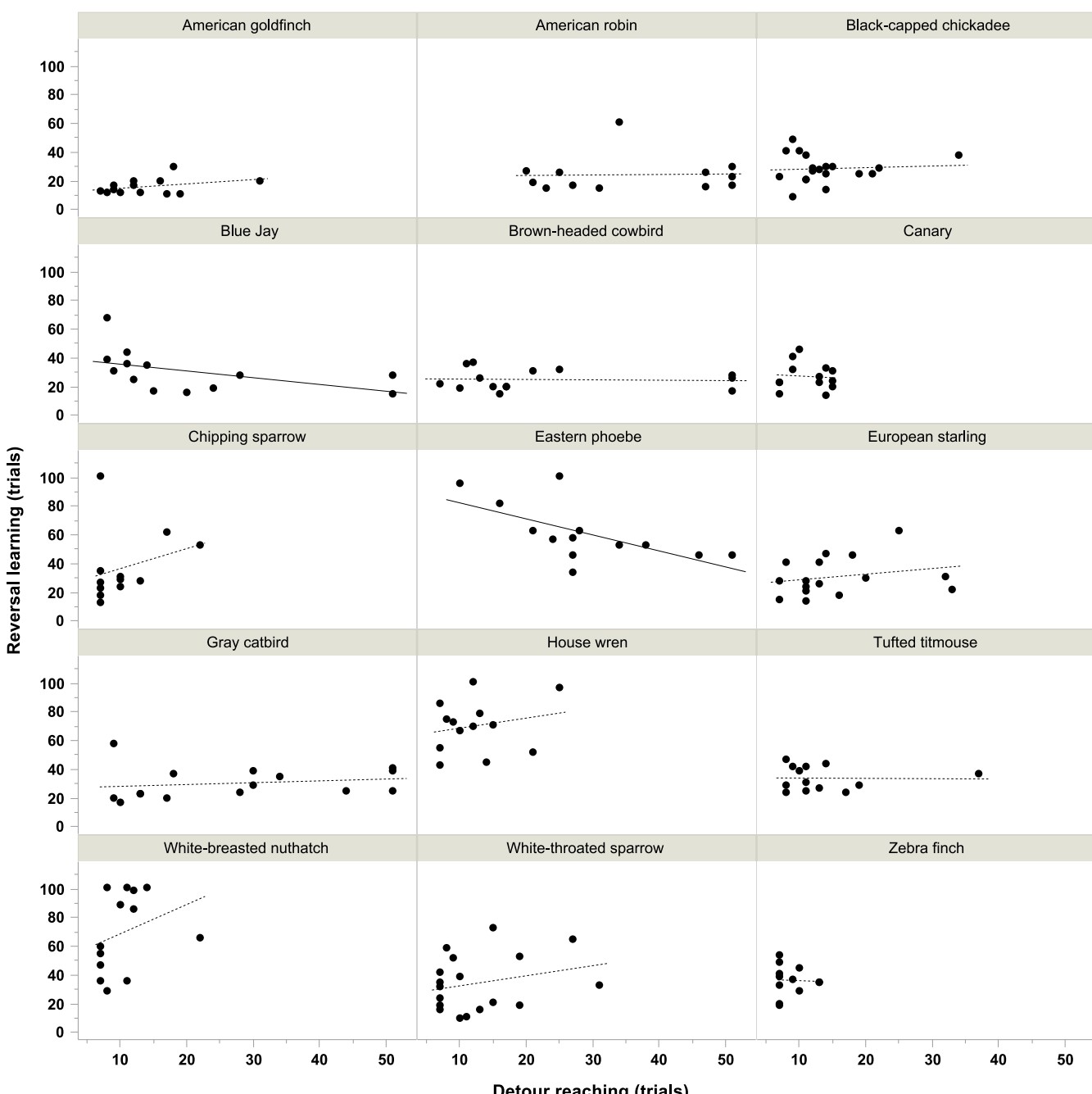

**Extended Data Fig. 7 | Intraspecific relationships between reversal learning and self-control performance for each of the 15 species.** Individual performance in the reversal learning task is significantly and negatively associated with detour reaching performance in the blue jay ($R^2$ = 0.351, $P$ = 0.0193), eastern phoebe ($R^2$ = 0.338, $P$ = 0.0219), but not in the other species (all other P > 0.05). $R^2$ and P-values were obtained from linear models (details in Supplementary Table 4e); solid regression lines: $P$ < 0.05, dashed regression lines: $P$ > 0.05.

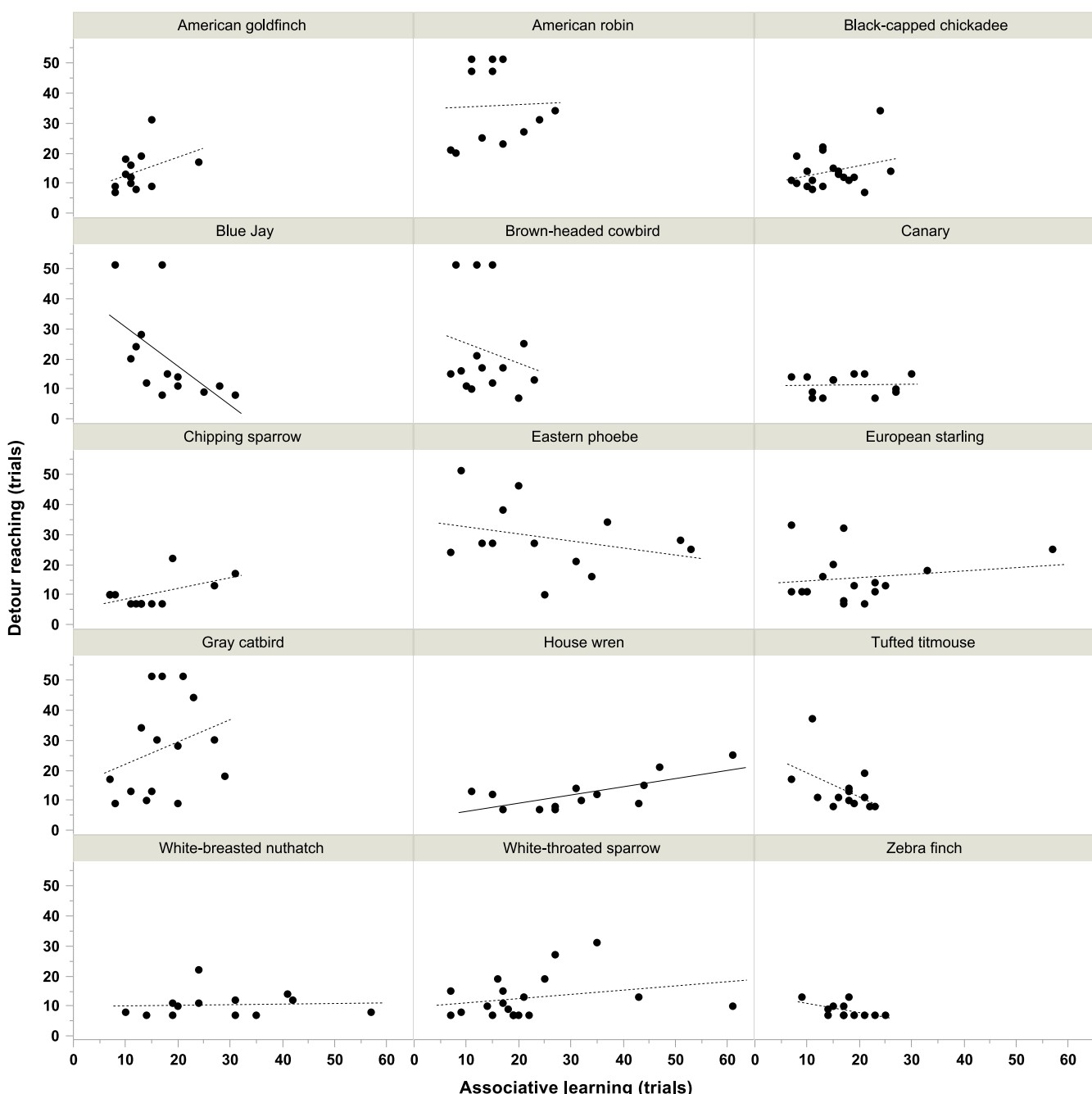

**Extended Data Fig. 8 | Intraspecific relationships between self-control and associative learning performance for each of the 15 species.** Individual performance in the detour reaching task is significantly and negatively associated with associative learning performance in the blue jay ($R^2 = 0.518$, $P = 0.0034$), positively in the house wren ($R^2 = 0.435$, $P = 0.0085$), but not in the other species (all other $P > 0.05$). $R^2$ and P-values were obtained from linear models (details in Supplementary Table 4f); solid regression lines: $P < 0.05$, dashed regression lines: $P > 0.05$.

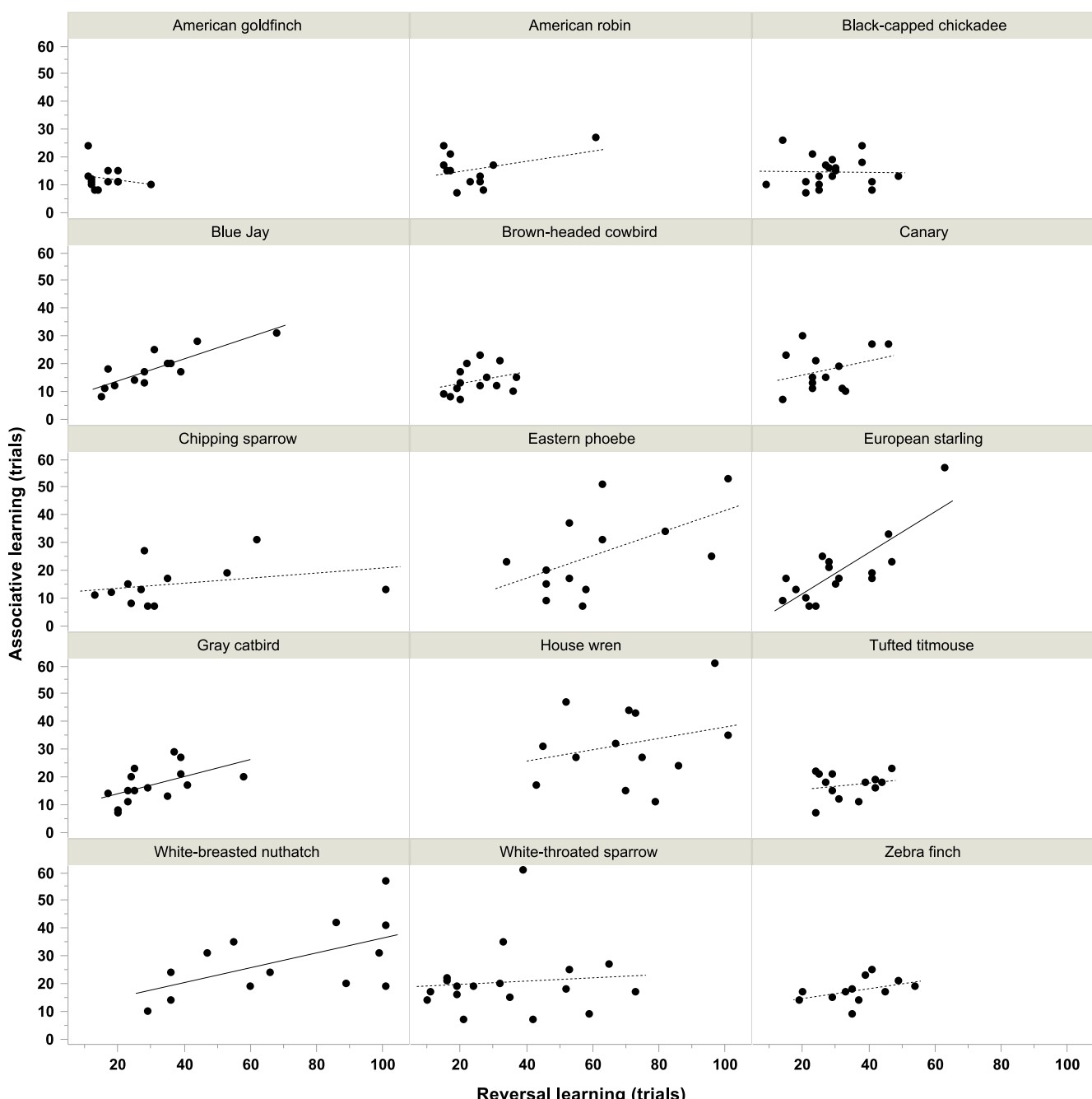

**Extended Data Fig. 9 | Intraspecific relationships between associative learning and reversal learning performance for each of the 15 species.** Individual performance in the associative learning task is significantly and positively associated with reversal learning performance in the blue jay ($R^2 = 0.691$, $P = 0.0003$), European starling ($R^2 = 0.631$, $P = 0.0001$), grey catbird ($R^2 = 0.229$, $P = 0.0406$) and white-breasted nuthatch ($R^2 = 0.260$, $P = 0.0431$), but not in the other species (all other $P > 0.05$). $R^2$ and P-values were obtained from linear models (details in Supplementary Table 4d); solid regression lines: $P < 0.05$, dashed regression lines: $P > 0.05$.

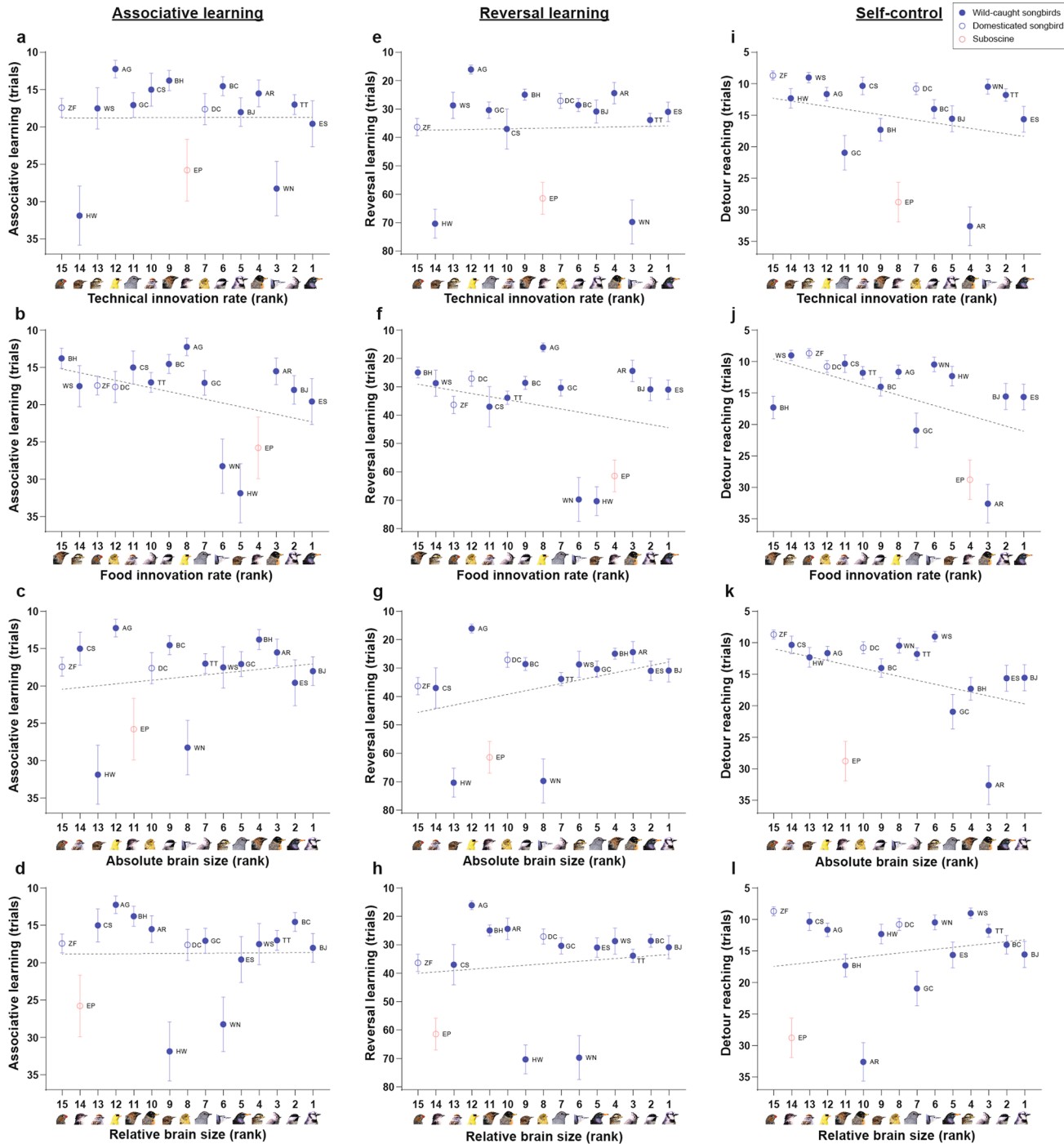

**Extended Data Fig. 10 | Relationships between species' performance on each cognitive task, innovation rates and brain size. a–d**, Associative learning performance between species is not significantly associated with their technical innovation rates (**a**), food innovation rates (**b**), absolute brain size (**c**) or relative brain size (**d**). **e–h**, Reversal learning performance between species is not significantly associated with their technical innovation rates (**e**), food innovation rates (**f**), absolute brain size (**g**) or relative brain size (**h**). **i–l**, Detour-reaching performance (self-control) between species is not significantly associated with their technical innovation rates (**i**), food innovation rates (**j**), absolute brain size (**k**) or relative brain size (**l**). Innovation values are residuals of innovation reports corrected for investigator research effort obtained from refs. 3,26; relative brain

sizes are the residuals of brain volumes corrected for average species' body weight, and absolute brain sizes are brain volumes; brain size and body weight data were obtained from ref. 27. Graphs illustrate mean species' trial values with s.e.m., ranked predictors and lines of values predicted by Bayesian phylogenetic mixed models. Filled blue circles, wild-caught songbird species; empty blue circles, domesticated songbird species (zebra finch and canary); red circles, the suboscine (eastern phoebe); dashed trend lines, $P_{\text{MCMC.adj}} > 0.05$; species' two-letter codes are listed in Supplementary Table 1; detailed results of MCMCglmm modelling and FDR-corrected $P$ values ($P_{\text{MCMC.adj}}$) are provided in Supplementary Table 2e–g. Image credits: Derrick Eidam for wild species and Mélanie Couture for domesticated species (zebra finch and canary).

# Reporting Summary

## Statistics

For all statistical analyses, confirm that the following items are present in the figure legend, table legend, main text, or Methods section.

| n/a | Confirmed | |
|---|---|---|
| ☐ | ☒ | The exact sample size (*n*) for each experimental group/condition, given as a discrete number and unit of measurement |
| ☒ | ☐ | A statement on whether measurements were taken from distinct samples or whether the same sample was measured repeatedly |
| ☐ | ☒ | The statistical test(s) used AND whether they are one- or two-sided<br>*Only common tests should be described solely by name; describe more complex techniques in the Methods section.* |
| ☐ | ☒ | A description of all covariates tested |
| ☐ | ☒ | A description of any assumptions or corrections, such as tests of normality and adjustment for multiple comparisons |
| ☐ | ☒ | A full description of the statistical parameters including central tendency (e.g. means) or other basic estimates (e.g. regression coefficient) AND variation (e.g. standard deviation) or associated estimates of uncertainty (e.g. confidence intervals) |
| ☐ | ☒ | For null hypothesis testing, the test statistic (e.g. *F*, *t*, *r*) with confidence intervals, effect sizes, degrees of freedom and *P* value noted<br>*Give P values as exact values whenever suitable.* |
| ☐ | ☒ | For Bayesian analysis, information on the choice of priors and Markov chain Monte Carlo settings |
| ☒ | ☐ | For hierarchical and complex designs, identification of the appropriate level for tests and full reporting of outcomes |
| ☐ | ☒ | Estimates of effect sizes (e.g. Cohen's *d*, Pearson's *r*), indicating how they were calculated |

*Our web collection on statistics for biologists contains articles on many of the points above.*

## Software and code

Policy information about availability of computer code

| | |
|---|---|
| Data collection | The collected behavioural dataset is available at: https://zenodo.org/records/10206756. Innovation and brain size data was obtained from published databases. |
| Data analysis | All analyses have been conducted in R version 4.3.0. The functions and packages we used are stated in the methods. |

For manuscripts utilizing custom algorithms or software that are central to the research but not yet described in published literature, software must be made available to editors and reviewers. We strongly encourage code deposition in a community repository (e.g. GitHub). See the Nature Portfolio guidelines for submitting code & software for further information.

## Data

Policy information about availability of data

All manuscripts must include a data availability statement. This statement should provide the following information, where applicable:

- Accession codes, unique identifiers, or web links for publicly available datasets
- A description of any restrictions on data availability
- For clinical datasets or third party data, please ensure that the statement adheres to our policy

Dataset and R code are available at https://zenodo.org/records/10206756.

## Human research participants

Policy information about studies involving human research participants and Sex and Gender in Research.

| | |
|---|---|
| Reporting on sex and gender | *Use the terms sex (biological attribute) and gender (shaped by social and cultural circumstances) carefully in order to avoid confusing both terms. Indicate if findings apply to only one sex or gender; describe whether sex and gender were considered in study design whether sex and/or gender was determined based on self-reporting or assigned and methods used. Provide in the source data disaggregated sex and gender data where this information has been collected, and consent has been obtained for sharing of individual-level data; provide overall numbers in this Reporting Summary. Please state if this information has not been collected. Report sex- and gender-based analyses where performed, justify reasons for lack of sex- and gender-based analysis.* |
| Population characteristics | *Describe the covariate-relevant population characteristics of the human research participants (e.g. age, genotypic information, past and current diagnosis and treatment categories). If you filled out the behavioural & social sciences study design questions and have nothing to add here, write "See above."* |
| Recruitment | *Describe how participants were recruited. Outline any potential self-selection bias or other biases that may be present and how these are likely to impact results.* |
| Ethics oversight | *Identify the organization(s) that approved the study protocol.* |

Note that full information on the approval of the study protocol must also be provided in the manuscript.

# Field-specific reporting

Please select the one below that is the best fit for your research. If you are not sure, read the appropriate sections before making your selection.

☐ Life sciences    ☐ Behavioural & social sciences    ☒ Ecological, evolutionary & environmental sciences

For a reference copy of the document with all sections, see nature.com/documents/nr-reporting-summary-flat.pdf

# Ecological, evolutionary & environmental sciences study design

All studies must disclose on these points even when the disclosure is negative.

| | |
|---|---|
| Study description | This study tested relationships between avian behavioural data collected in the field and literature data on innovation and brain size. |
| Research sample | We collected behavioural data on 203 individuals of 13 wild and 2 domesticated avian species. |
| Sampling strategy | The species were chosen based on their abundance where the study was conducted (Rockefeller Field Research Center). We aimed at reaching a sample size of >= 12 individuals per species to account for individual variation in the behaviour we measured. We expected that 15 species would be sufficient to test for associations with species-specific data on innovation and brain size. |
| Data collection | The behavioural data were collected by JNA, by observing captive birds performing on our battery of behavioural tasks. |
| Timing and spatial scale | The field seasons occured from 2018 to 2020. |
| Data exclusions | No data were excluded from the analyses. |
| Reproducibility | The same behavioural protocol was used for all individual of each species (total: 203 birds). |
| Randomization | The birds were tested in order of their capture. The behavioural tasks were not randomised since the test order is expected to strongly influence the performance; therefore, it was kept identical for all tested birds. |
| Blinding | The behavioural data were only analysed at the end of the field seasons; therefore, the results were unknown throughout the testing period. |

Did the study involve field work?    ☒ Yes    ☐ No

## Field work, collection and transport

| | |
|---|---|
| Field conditions | Birds were captured in any weather condition. When conditions were hostile (e.g., raining), mist nets were visited more often. |

| Location | Rockefeller Field Research Center (Millbrook, NY, USA, 41° 46′ 3.0″ N, 73° 45′ 2.5″ W) |
|---|---|
| Access & import/export | Field work was conducted in compliance with all local and national regulations. Permits were issued by Rockefeller University (IACUC permit # 17084), New York State Department of Environmental Conservation (Banding permit # 198, Scientific collection permit # 2284), United States Fish and Wildlife Service (Permit # MB-45822C) and United States Geological Survey (Permit # 24130). |
| Disturbance | We collected only the number of birds necessary to obtain a sufficient sample size to conduct our analyses. Non-target species were immediately released. |

# Reporting for specific materials, systems and methods

We require information from authors about some types of materials, experimental systems and methods used in many studies. Here, indicate whether each material, system or method listed is relevant to your study. If you are not sure if a list item applies to your research, read the appropriate section before selecting a response.

## Materials & experimental systems

| n/a | Involved in the study |
|---|---|
| ☒ | Antibodies |
| ☒ | Eukaryotic cell lines |
| ☒ | Palaeontology and archaeology |
| ☐ ☒ | Animals and other organisms |
| ☒ | Clinical data |
| ☒ | Dual use research of concern |

## Methods

| n/a | Involved in the study |
|---|---|
| ☒ | ChIP-seq |
| ☒ | Flow cytometry |
| ☒ | MRI-based neuroimaging |

## Animals and other research organisms

Policy information about studies involving animals; ARRIVE guidelines recommended for reporting animal research, and Sex and Gender in Research

| Laboratory animals | We collected behavioural data on canaries (Serinus canaria) and zebra finches (Taeniopygia guttata). Zebra finches were obtained from our domestic colony and canaries were bought from a local breeder. |
|---|---|
| Wild animals | We captured and collected behavioural data on the following species: White-throated sparrow (Zonotrichia albicollis); Chipping sparrow (Spizella paserina); Brown-headed cowbird (Molothrus ater); American goldfinch (Spinus tristis); American robin (Turdus migratorius); European starling (Sturnus vulgaris); Gray catbird (Dumetella carolinensis); House wren (Troglodytes aedon); White-breasted nuthatch (Sitta carolinensis); Black-capped chickadee (Poecile atricapillus); Tufted titmouse (Baeolophus bicolor); Blue Jay (Cyanocitta cristata); and Eastern phoebe (Sayornis phoebe). Birds were captured using mist nets and were brought in behaviour cages immediately. Except for a few birds that were sacrificed for another study, birds were released after the behavioural tests at their initial capture site. |
| Reporting on sex | To minimise the sample size, we used only males since the sex can potentially affect behavioural measures. We added females of two species (blue jay and European starling) because reaching a sufficient sample size of only males for these species proved to be challenging. The effect of sex was assessed for these species. |
| Field-collected samples | Birds were housed in an aviary kept at 70 degrees F. Lighting period was adjusted daily to reflect the natural photoperiod to minimise the stress on wild animals. |
| Ethics oversight | All procedures were approved by Rockefeller University, New York State Department of Environmental Conservation, United States Fish and Wildlife Service and United States Geological Survey. |

Note that full information on the approval of the study protocol must also be provided in the manuscript.

