## [Peer Review File · Nature Ecology & Evolution]

Peer Review Information

Journal: Nature Ecology & Evolution

Manuscript Title: Problem-solving skills predict innovations in the wild and brain size in Passerines

Corresponding author name(s): Jean-Nicolas Audet

Editorial Notes:

Reviewer Comments & Decisions:

Decision Letter, initial version:

5th May 2023

Dear Dr Audet,

Your manuscript entitled "Problem-solving skills predict innovations in the wild and brain size in Passerines" has now been seen by 3 reviewers, whose comments are attached. The reviewers have raised a number of concerns which will need to be addressed before we can offer publication in Nature Ecology & Evolution.

As you will see from the comments below, the reviewers have found your work to be of considerable interest and significance. However, they have raised common concerns regarding statistical tests and improvements to figure legends. Reviewer 3 has also pointed out potential issues with the measures used for analysis. We will therefore need to see your responses to the criticisms raised and to some editorial concerns, along with a revised manuscript, before we can reach a final decision regarding publication.

We therefore invite you to revise your manuscript taking into account all reviewer and editor comments. Please highlight all changes in the manuscript text file.

- * Include a "Response to reviewers" document detailing, point-by-point, how you addressed each reviewer comment. If no action was taken to address a point, you must provide a compelling argument. This response will be sent back to the reviewers along with the revised manuscript.
- * If you have not done so already please begin to revise your manuscript so that it conforms to our Article format instructions at <http://www.nature.com/natecolevol/info/final-submission>. Refer also to any guidelines provided in this letter.
- * Include a revised version of any required reporting checklist. It will be available to referees (and,

2potentially, statisticians) to aid in their evaluation if the manuscript goes back for peer review. A revised checklist is essential for re-review of the paper.

[REDACTED]

Nature Ecology & Evolution is committed to improving transparency in authorship. As part of our efforts in this direction, we are now requesting that all authors identified as 'corresponding author' on published papers create and link their Open Researcher and Contributor Identifier (ORCID) with their account on the Manuscript Tracking System (MTS), prior to acceptance. ORCID helps the scientific community achieve unambiguous attribution of all scholarly contributions. You can create and link your ORCID from the home page of the MTS by clicking on 'Modify my Springer Nature account'. For more information please visit www.springernature.com/orcid.

[REDACTED]

Reviewer expertise:

Reviewer #1:

Reviewer #2:

Reviewer #3:

Reviewers' comments:

Reviewer #1 (Remarks to the Author):

2In this paper Jean-Nicolas Audet and coworkers investigate how different cognitive tests relate to innovativeness in the wild and brain size across several bird species. The authors show that only the results of problem-solving tests are associated with innovativeness and brain size. I found the reasoning clear and the paper well-written. On the other hand, I have some concern about the statistical analyses performed in the paper.

First, the authors present the relationships among different tests, innovativeness and brain size across species in a few figure (e.g. Ext. Fig. 2, Fig. 3, Fig. 4). In these figures they use Spearman rank correlation to quantify the strength of the relationships. This would be a reasonable choice for these kind of data, but this approach fails to account for the phylogenetic relationship among the studied species. It is true that later the authors present a series of complicated analyses where they control for phylogeny, but that a different story because there they analyse individuals whereas here the focus is on species. It would be nice to see that these results remain after phylogenetic control included. A further problem with these figures is that the authors show regression lines in the panels, I guess to illustrate the relationships. But no details are given about these regressions. Furthermore, it is misleading to mix the results of correlation and regression analyses. Therefore I suggest to remove these lines from the figures.

Second, the MCMCglmm analyses are not entirely clear. What is the reason to analyse individual data in the case where variables of main interest (innovativeness, brain size) are only measured at the species level? Phylogeny can be controlled in species level models as well. Does the random factor Species refer to the phylogeny entered into the model? If not, why do you need species apart from phylogenetic control? What does ID refer to? Is the ID of individuals tested? But as far as I understand you have only one measurement for each bird in each variable. Where you have more than one measurement, like shyness, or problem-solving you took the averages of these. The understanding of these analyses can significantly be helped by making the code of the analyses available.

Third, the authors perform many statistical tests but it is unclear whether they used p-values adjusted for multiple comparison. On line 96 they state that in certain cases they do so, but what's about the other cases, e.g. in Table 1.

L80-81: "Extended Data Movie 5" should be "Extended Data Movie 7"

L85: "Movies 6-7" should be "Movies 5-6"

L120: Shouldn't "associative learning" be listed here as well?

L546: "task" -> "flask"?

Reviewer #2 (Remarks to the Author):

The study uses an array of laboratory tasks (problem solving, associative learning, reversal learning, and self-control) to measure cognition in 203 birds from 15 passerine species. A key finding is that

3performance on one task is generally unrelated to performance on another, both within and among species. This finding helps resolve a long-standing debate about whether cognition is modular or whether different aspects of cognition can be subsumed under the broader concept of behavioural flexibility. A second key finding is that performance on the problem-solving task predicts published estimates of a species' brain size and innovation rate in the wild, even after controlling for phylogeny, personality, and other variables that might influence the relationship. This study is the first that I am aware of that links a species' brain size and innovation rate to its performance on standardized cognitive tasks. This finding is exciting because innovation rate in the wild is a critical determinant of a species' extinction risk and ability to colonize new environments. The finding therefore will be of considerable interest to several fields, including cognition, animal behaviour, urban ecology, and conservation. The study represents a formidable effort, and the data set is impressive in terms of sample size, species representation, and the comprehensive assessment of cognition. I enjoyed reviewing the manuscript and hope that my feedback proves useful. Sincerely, Dave Wilson

General comments:

I appreciate the general statistical approach of using simple correlations and scatterplots to show main effects, and then using phylogenetically controlled models to rule out alternative explanations. Part of the value of using correlations is that, for most analyses, there is no logical assignment of variables as independent and dependent (e.g., innovation rate vs. problem solving score). The MCMCGLMMs, however, make these assumptions by assigning some variables as dependent and others as independent. This is probably unavoidable because these models require dependent and independent variable assignments, but the choice of these assignments sometimes contradicts the message of the paper. For example, the title states that problem-solving skills PREDICT innovations and brain size, yet language throughout the manuscript (e.g., L103, 162-163) and the MCMCGLMMs (L126-146, 638-651, Table 1, extended data table 3) contradict this by treating brain size and innovation as the predictors (independent variables) of problem solving. One solution would be to make innovation rate and brain size the dependent variables and include the cognitive test scores as predictor variables. Using this approach, it might be possible to include neophobia, shyness, problem solving, associative learning, reversal learning, and self-control together as predictors in the same models, which would better support statements of the relative effects of these different cognitive processes (e.g., L186-190). Separate models would then be run with total innovations, food innovations, technical innovations, relative brain size, and absolute brain size as dependent variables. I don't believe any of the main findings would change with this approach. At a minimum, the directionality implied the word 'predict' needs to be consistent throughout the manuscript and needs to match the directionality assumed by the assignment of independent and dependent variables in the MCMCGLMMs.

The title, abstract, introduction, results, and discussion imply that the results apply broadly to all passerines, but, except for 7 (out of 203) individuals from 2 (out of 15) species (L414-417), all subjects were male. The results therefore cannot be generalized to both sexes, and the paper should acknowledge this more explicitly throughout.

I wonder if the figures and tables could be enriched with species information so that readers can see how the species are distributed across a given variable. I suggest numbering the data points in figures 3 and 4 (though the heads along the x-axis in figure 4 help) and extended figures 2, 3, and 10, and

4then linking them to species names in table 1 (by adding a new column to indicate the species' reference number) and figure 2 (by providing the number with the species name on the phylogeny) (and/or to a key in the caption of each figure). Perhaps this will make the figures too cluttered, but, if not, the extra information would be very helpful.

Specific comments:

L43-45: provide a few words to help the reader visualize what you mean by a self-control task, associative learning task, and reversal task, as has been done for problem-solving tasks. I believe the study will be of interest to researchers from outside cognition, so it would be helpful if they could visualize these tasks earlier in the introduction.

L58: it would be useful to include a sentence in the introduction to list some of the factors that have been linked to avian cognition in other studies (e.g., urbanization), since this will provide rationale for the covariates included (or excluded) in the current study.

L71: 967 out of how many? By 'cases', do you mean species or examples of innovations?

L106: for non-specialists, it would be helpful to include an example or brief description of 'innovations' to help them visualize what is being tested.

L117-118: a very significant finding!

L119-125: should this section not also include 'associative learning'?

L138: the clause beginning '; and body condition...' sounds awkward and should be revised

L188-190: should specify here which assay tests which of the cognitive processes mentioned here; also, what about the third assay used in the study - should it also be mentioned here; similarly, should associative learning be included on L196?

L402: please provide range of dates per year when the trials were conducted, and whether trials were confined to the breeding season when male birds are easier to capture.

L425-427: should clarify whether birds were housed individually or socially in each cage

L459: should note here that birds were tested in their home cages

L550-551: should specify how many birds were successful on the confirmation trial.

L569: should this say 'day 2' rather than 'day 1' to match L552?

L595: if a bird chose the rewarded colour in trials 1-7, would they be given a score of 1, 7, or 3.5. Should clarify scoring scheme.

L618: typo 'residues'; also, is the relationship between # innovations and research effort necessarily linear? I would think that # of innovations would plateau with increasing effort, though perhaps few species have sufficient research effort to reach that point.

L622: where did the body mass data come from, and were they specific to males given that most birds in the study were male?

L643-645: should specify how many categories there were for 'capture site,' since I thought all wild birds were captured at the same site (Rockefeller Field Station); also, why was shyness included in the models for neophobia, but neophobia was not included in the models for shyness (extended Table 3); this sentence should also include 'dietary generalism', and another sentence should explain where those data came from

L645: typo - change MGMG to MCMC

L664: unclear what is meant by 'excluding seasonal migrants'. Aren't several of your species migratory at your study site (e.g., goldfinch, cowbird, sparrows, phoebe, catbird, wrens)?

Extended Fig. 2 caption: tasks are described as 'lid-pulling' and 'lid-flipping', but as 'lid-knocking' and 'lid-flipping' elsewhere

Extended Fig. 3: fig caption uses directional language, such as 'species that perform better on problem-solving tasks are not shy or more neophobic, but the basis of those directional predictions is unclear. For example, why couldn't birds with better problem solving be bolder rather than shy? If the directionality is to be maintained, rationale should be provided, and the correlation tests should be 1-tailed. I recommend changing the wording here to say that trait X was not related to trait Y.

Extended Table 1: please add a column for family to help readers from outside ornithology

Extended Table 3: in model 25a, the line for 'reward' should be in bold

Videos: very helpful, but the versions I had for review were quite grainy; the two showing the self-control test, in particular, were difficult to see what was happening; if higher-quality versions exist, they would be helpful

Reviewer #3 (Remarks to the Author):

The study uses an impressive range of cognitive tests comparing over 200 individuals across 15 songbird species. This gives depth to the findings regarding a more general concept of factors affecting innovation rates. The study is well executed and written. However, I have some concerns about the calculation of 'pure neophobia', which should be revisited. Moreover, I wondered whether any corrections were used for the many Spearman correlations. If not, this should be added.

6Main:

Relevant background information has been provided that leads to the aims. The approach is innovative and one of very few addressing all three steps usually linked to innovation across a good range of songbird species. The paper is relevant for psychologists, behavioural ecologists and cognitive ecologists, and therefore of interest across disciplinary areas.

Recommendation:

Key results: The study investigated several cognitive abilities often linked to innovation using an extensive set of tests in a large sample covering 15 songbird species to establish the underlying factors affecting innovations in the wild. The key finding is that problem-solving abilities in captivity seem to be the best predictor of innovation rates in the wild also correlating with brain size, whereas other factors showed little relationship to innovation rate or brain size. Given the breadth of experiments conducted and the number of individuals and species tested, this constitutes a major step forward in explaining innovation rates in birds.

Validity: Results are well supported by large sample sizes and thorough testing. However, I am concerned that the 'pure neophobia' variable is not representative of neophobia as it is currently calculated (see comment below regarding lines 521-522 and 529-530). Moreover, I could nowhere find that there were any corrections used for multiple testing (Spearman correlations; see comment regarding e.g., lines 633-637). Finally, the authors may consider testing individuals nested within species (see comment regarding lines 643-645). All issues might affect the outcome but can be addressed in a revision.

Originality and significance: Conclusions are original and provide an important contribution to increase knowledge about factors affecting innovation. The large range of tests used in combination with a large sample size covering 15 songbird species makes this a robust and significant finding. The design is likely to be adopted by other researchers to test similar relationships in other taxa and is therefore a large relevance. The study combined psychological approaches with ecological theory and is of relevance to a broad readership across disciplines.

Data & methodology: Data belong to over 200 individuals of 15 songbird species which is an impressive sample in this field of research. The methodology is thorough and uses an extensive set of tests that surpasses most other studies. Therefore, the data are of high quality. Methods are clearly described and established procedure have been used (but see comments above regarding neophobia). Appropriate use of statistics and treatment of uncertainties: Appropriate tests have been used (but see comment about corrections for multiple testing). Legends need in part more information (see comments regarding Extended Data Fig. 2 and 10 and main Figs. 3 and 4).

Conclusions: Conclusions are valid and robust and reflect the results.

Suggested improvements:

Fig. 1: The figure provides an informative overview about factors affecting innovations and tests used. However, other factors have been ignored. For example, while dietary generalism has been linked to innovations, persistence has also been identified (Griffin & Guez 2014. *Behavioural Processes*. 109:121–134. doi:10.1016/j.beproc.2014.08.027.) and should be included here. Moreover, shyness and neophobia have been listed to potentially affect problem-solving abilities. One important factor is missing here as exploration (neophilia) has been repeatedly linked to problem-solving abilities (e.g., Griffin & Guez 2014). It should be mentioned that neophobia and neophilia are not the extremes along a continuum but represent independent motivations (see 2-Factor model in Greenberg & Mettke-Hofmann 2002, *Ethology* 108, 249-272). This should be added in the text. One other point: it has nowhere been explained what the solid and hatched lines mean.

Results:

Extended Data Fig. 2 and main Figs. 3 and 4: *, ** and *** are not explained in the legend. R and P values are from Spearman correlations. This required many tests. Were there any corrections included, e.g., sequential Bonferroni tests to account for the number of tests performed? If so, this should be mentioned in the legend and the methods section. If no corrections have been done, this would be an important aspect to add to reduce type 1 errors (this applies to all figures and analyses).

Extended Data Fig. 10: The y-axis label is missing.

Extended Data Table 3: While the text mentions that interactions were considered, there is not a single interaction listed in the table. This is odd as it is difficult to imagine that there are no significant interaction terms. Can this please be confirmed, or meaningful interactions be added in the model.

Lines 143-146: I would not say that some of the relationships were weak associations as problem-solving and shyness for example were consistently highly significant. On what basis do you judge a relationship weak or strong?

Discussion:

Lines 148pp: While I agree that most other cognitive traits were not related to innovation rate or brain size, I think the consistent link between problem-solving ability and shyness needs mentioning and discussion.

Lines 513-517: For shyness, only the first latency to feed without anything next to the food was measured on day one. Shyness is a personality measure, and an important characteristic of personality is its consistency over time (e.g., Wilson et al. 1994; TREE 9(11), 442-446). However, this consistency has nowhere been shown. The authors report a decline in response times over time (habituation). They could use the same data to test for consistency in responses on the individual level, i.e., are individuals that were slow to feed on the first day still slow to feed on the e.g., last day of testing relative to the other birds. Otherwise, this measure should not be called shyness, particularly as the used measure is quite unusual.

Lines 521-522: Why did you set neophobia to zero when the shyness measure was higher than the neophobia measure? The correct way would be to use the correct negative value. Martin II & Fitzgerald (2005; Behavioral Ecology, doi:10.1093/beheco/ari044) for example reported a faster approach to food with novel objects present and interpreted this as attraction to novelty (neophilia). I suggest repeating the analyses with the correct values.

Lines 529-530: I wonder whether the average across four days of neophobia testing is a good idea. Your control variable 'shyness' decreased over the four testing days, whereas the time to feed with the novel objects might have stayed the same (one would not expect habituation of neophobia towards different objects). This would increase pure neophobic responses from day 1 to 4 due to the decrease in shyness (you can test for this and when this is indeed the case your pure neophobia measure using the average across the four days is not a true reflection of an individuals/species' neophobia and should not be used). This reflects one of the problems using shyness as the control as it is not a consistent representation how fast an individual usually feeds day in and day out. A better way would have been to give the birds more time to habituate and wait until they consistently feed without objects around. This value could then have served as a control measure for neophobia. At the moment, I cannot see how to solve this issue. The first pure neophobia measure is unlikely the correct neophobia response and likely the lowest neophobia among the four measures due to the high shyness measure, i.e., shyness and neophobia are likely to interact here. The last pure neophobia measurement is possibly the best representation of neophobia as the birds have habituated to the experiment and shyness does not interact with neophobia anymore. Maybe just using this last

measure would be a better representation of neophobia than the average across the four days unless you can show that pure neophobia responses were highly correlated within individuals across the four days. Another option could be to use the pure neophobia measure of each day for the respective cognitive test, but I assume this would be very difficult to handle. In any way, one or the other alternative should be explored and tested to see whether it has any effect on the results. I would expect that any potential effect of neophobia on problem-solving and/or innovation would increase. Lines 633-637: Many Spearman correlations were used resulting in type 1 errors. Were any corrections used (e.g., sequential Bonferroni adjustments) to account for this? If not, this should be done.

Lines 643-645: Species were included as random factors. This accounts for repeated testing of the same species. However, shouldn't there be a nested approach be used with testing individuals within species to account for closely related individuals (e.g., belonging to the same species) to be more similar in their responses?

References: Relevant references are included.

Clarity and context: Abstract and main text are well written.

*****END*****

Author Rebuttal to Initial comments

Reviewer #1 (Remarks to the Author):

In this paper Jean-Nicolas Audet and coworkers investigate how different cognitive tests relate to innovativeness in the wild and brain size across several bird species. The authors show that only the results of problem-solving tests are associated with innovativeness and brain size. I found the reasoning clear and the paper well-written. On the other hand, I have some concern about the statistical analyses performed in the paper.

Response: We thank the reviewer for the positive comments. We address the specific concerns below.

First, the authors present the relationships among different tests, innovativeness and brain size across species in a few figure (e.g. Ext. Fig. 2, Fig. 3, Fig. 4). In these figures they use Spearman rank correlation to quantify the strength of the relationships. This would be a reasonable choice for these kind of data, but this approach fails to account for the phylogenetic relationship among the studied species. It is true that later the authors present a series of complicated analyses where they control for phylogeny, but that a different story because there they analyse individuals whereas here the focus is on species. It would be nice to see that these results remain after phylogenetic control included. A further problem with these figures is that the authors show regression lines in the panels, I guess to illustrate the relationships. But no details are given about these regressions. Furthermore, it is misleading to mix the results of correlation and regression analyses. Therefore I suggest to remove these lines from the figures.

Response: The reviewer is correct that our initial correlation analyses are without phylogenetic correction, which we purposely have done so. We think it is important for the readers to see the raw relationships before other types of potential controls. We followed the reviewer's suggestion for phylogenetic correction analyses by further implementing phylogenetic distance correction for all between-species tests (see the answer to the reviewer's following comment).

In terms of mixing results of correlation and regression, we agree that if not explained well, it can be misleading. But, adding regression lines on correlation graphs to visualise the strength of relationships is commonly done. We now provide more details about the presented p-values and regression lines in the figure legends: "R and P are obtained from Spearman correlations. (...); lines were computed using linear regressions on the same variables to illustrate the strength of relationships". We did remove the 95% confidence intervals, as the statistical values shown were not based on such analyses.

Second, the MCMCglmm analyses are not entirely clear. What is the reason to analyse individual data in the case where variables of main interest (innovativeness, brain size) are only measured at the species level? Phylogeny can be controlled in species level models as well. Does the random factor Species refer to the phylogeny entered into the model? If not, why do you need species apart from phylogenetic control? What does ID refer to? Is the ID of individuals tested? But as far as I understand you have only one measurement for each bird in each variable. Where you have more than one measurement, like shyness, or problem-solving you took the averages of these. The understanding of these analyses can significantly be helped by making the code of the analyses available.

Response: We performed MCMCglmm full models to control for other potential factors measured at the individual level. As the reviewer suggests, our full modelling MCMCglmm analyses control for between species (n = 15) phylogenetic relationships, but also individual behavioural variation (n = 203 total) within each species. Following the reviewer's comment, we have performed an additional MCMCglmm, controlling for between-species phylogenetic relationship only (n = 15) in interspecific tests, without co-modelling other control variables among individuals. The main conclusions of relationships between problem-solving and technical innovation or brain size remained unchanged (added into Extended Data Tables 2 and 3).

We acknowledge that the variable name "ID" was misleading. This is meant not for the ID of the individual, but for the ID of the species (now termed "species" in model outputs).

The code is now available here: <https://zenodo.org/record/8190695>

Third, the authors perform many statistical tests but it is unclear whether they used p-values adjusted for multiple comparison. On line 96 they state that in certain cases they do so, but what's about the other cases, e.g. in Table 1.

Response: Considering the reviewer's comment, we now provide corrected p-values for all statistical tests (Spearman correlations, MCMCglmm with species means, final MCMCglmm with individual variation in Table 1). We used false discovery rate (FDR) Benjamini & Hochberg p-value correction. While the corrected p-values differ, most conclusions remain, except for the relationship between problem-solving and total innovation when correcting for phylogenetic relationships. We now mention this new result in the manuscript [L131-133]. The adjustment procedure has been added in the methods [L717-722].

L80-81: "Extended Data Movie 5" should be "Extended Data Movie 7"

L85: "Movies 6-7" should be "Movies 5-6"

L120: Shouldn't "associative learning" be listed here as well?

L546: "task" -> "flask"?

Response: We thank the reviewer for catching the above typos; we corrected them.

Reviewer #2 (Remarks to the Author):

The study uses an array of laboratory tasks (problem solving, associative learning, reversal learning, and self-control) to measure cognition in 203 birds from 15 passerine species. A key finding is that performance on one task is generally unrelated to performance on another, both within and among species. This finding helps resolve a long-standing debate about whether cognition is modular or whether different aspects of cognition can be subsumed under the broader concept of behavioural flexibility. A second key finding is that performance on the problem-solving task predicts published estimates of a species' brain size and innovation rate in the wild, even after controlling for phylogeny, personality, and other variables that might influence the relationship. This study is the first that I am aware of that links a species' brain size and innovation rate to its performance on standardised cognitive tasks. This finding is exciting because innovation rate in the wild is a critical determinant of a species' extinction risk and ability to colonise new environments. The finding therefore will be of considerable interest to several fields, including cognition, animal behaviour, urban ecology, and conservation. The study represents a formidable effort, and the data set is impressive in terms of sample size, species representation, and the comprehensive assessment of cognition. I enjoyed reviewing the manuscript and hope that my feedback proves useful. Sincerely, Dave Wilson

Response: We are thankful for this appreciation of our work.

General comments:

I appreciate the general statistical approach of using simple correlations and scatterplots to show main effects, and then using phylogenetically controlled models to rule out alternative explanations. Part of the value of using correlations is that, for most analyses, there is no logical assignment of variables as independent and dependent (e.g., innovation rate vs. problem solving score). The MCMCGLMMs, however, make these assumptions by assigning some variables as dependent and others as independent. This is probably unavoidable because these models require dependent and independent variable assignments, but the choice of these assignments sometimes contradicts the message of the paper. For example, the title states that problem-solving skills PREDICT innovations and brain size, yet language throughout the manuscript (e.g., L103, 162-163) and the MCMCGLMMs (L126-146, 638-651, Table 1, extended data table 3) contradict this by treating brain size and innovation as the predictors (independent variables) of problem solving. One solution would be to make innovation rate and brain size the dependent variables and include the cognitive test scores as predictor variables. Using this approach, it might be possible to include neophobia, shyness, problem solving, associative learning, reversal learning, and self-control together as predictors in the same models, which would better support statements of the relative effects of these different cognitive processes (e.g., L186-190). Separate models would then be run with total innovations, food innovations, technical innovations, relative brain size, and absolute brain size as dependent variables. I don't believe any of the main findings would change with this approach. At a minimum, the directionality implied the word 'predict' needs to be consistent throughout the manuscript and needs to match the directionality assumed by the assignment of independent and dependent variables in the MCMCGLMMs.

Response: We agree that the title contradicted our modelling that assigned innovation and brain size as predictors of problem-solving. We believe our modelling design is appropriate; therefore, we modified the title and main text to reflect this. Although it is difficult to determine causality or

directionality between our cognitive measurements and published values of innovation or brain size, we find it more logical to assign the experimental measures as the dependent variables in our models and published metrics as predictors. Finally, to assess the effect of potential covariates on our cognitive measures, the cognitive measures need to be placed as dependent variables in models, and the covariates as independent (e.g., having both problem-solving and shyness placed as independent covariables would introduce autocorrelations problems). Therefore, we kept our modelling strategy the same but modified the title and all instances in the manuscript where directionality was implied.

The title, abstract, introduction, results, and discussion imply that the results apply broadly to all passerines, but, except for 7 (out of 203) individuals from 2 (out of 15) species (L414-417), all subjects were male. The results therefore cannot be generalised to both sexes, and the paper should acknowledge this more explicitly throughout.

Response: We added a mention of most individuals being males in the main manuscript [L63-65]. We also mention that for those species for which we had both sexes, we found no significant behavioural differences between males and females (L454-458 and Extended Data Table 6). We believe it's very unlikely that if we had measured subtle differences between males and females in other species, it would have influenced our discovered relationships with species-specific metrics of innovation and brain size.

I wonder if the figures and tables could be enriched with species information so that readers can see how the species are distributed across a given variable. I suggest numbering the data points in figures 3 and 4 (though the heads along the x-axis in figure 4 help) and extended figures 2, 3, and 10, and then linking them to species names in table 1 (by adding a new column to indicate the species' reference number) and figure 2 (by providing the number with the species name on the phylogeny) (and/or to a key in the caption of each figure). Perhaps this will make the figures too cluttered, but, if not, the extra information would be very helpful.

Response: We agree with the reviewer's point. Instead of adding numbers, we added species two-letter codes in all graphs (the code reference has been added in Extended Data Table 1).

Specific comments:

L43-45: provide a few words to help the reader visualise what you mean by a self-control task, associative learning task, and reversal task, as has been done for problem-solving tasks. I believe the study will be of interest to researchers from outside cognition, so it would be helpful if they could visualise these tasks earlier in the introduction.

Response: We added a short description for self-control and associative/reversal learning tasks [L44-48]: "(...) *self-control tasks, which measure the ability to inhibit a prepotent but unproductive behaviour, e.g., to find an alternate route to obtain a reward, assess inhibitory control*¹³; *puzzle boxes requiring extractive foraging and obstacle removal, sometimes with tools, assess novel problem-solving*¹⁴; and *association and reversal tasks, which measure the ability to discriminate between rewarded and unrewarded cues, target the efficiency of learning of new cues*¹⁵".

L58: it would be useful to include a sentence in the introduction to list some of the factors that have been linked to avian cognition in other studies (e.g., urbanisation), since this will provide rationale for the covariates included (or excluded) in the current study.

Response: Following the reviewer's request, we added the following sentences: *"for example, when colonising new areas, including cities. Consistent with this notion, problem-solving speed is positively associated with urbanisation degree (e.g., ^{10,17}) and consumption of anthropogenic food¹⁸. However, associative and reversal learning speed has been found to correlate with urbanisation negatively^{19,20"} [L50-53].*

L71: 967 out of how many? By 'cases', do you mean species or examples of innovations?

Response We added "967 cases of innovation (out of 4452)" [L81-82].

L106: for non-specialists, it would be helpful to include an example or brief description of 'innovations' to help them visualise what is being tested.

Response: We added a few words in the main section [L118], and a short description in the methods section [L646-661], which now reads *"Innovations are published cases of novel feeding (incorporation of an unusual or previously unknown food source in the animal's diet) or technical (use of a novel foraging technique) behaviours, in the literature, based on the presence in the report of key words like "new", "never observed", "first report", "not mentioned in the literature", "opportunistic", etc. "*

L117-118: a very significant finding!

L119-125: should this section not also include 'associative learning'?

Response: We added associative learning as suggested.

L138: the clause beginning '; and body condition...' sounds awkward and should be revised

Response: We believe using colons and semi-colons inside the parentheses made this long sentence awkward. We simplified the sentence and removed colons.

L188-190: should specify here which assay tests which of the cognitive processes mentioned here; also, what about the third assay used in the study - should it also be mentioned here; similarly, should associative learning be included on L196?

Response: We added the specific assays [L212-213] and "associative learning" [L219].

L402: please provide range of dates per year when the trials were conducted, and whether trials were confined to the breeding season when male birds are easier to capture.

Response: We added the months of captures [L439] which included, but were not entirely limited to, the breeding seasons.

L425-427: should clarify whether birds were housed individually or socially in each cage

Response: We added "individually" [L465] and also, "The birds were visually but not acoustically isolated from each other" [L467-468].

L459: should note here that birds were tested in their home cages

Response: We added birds were tested "in their home cages" [L499].

L550-551: should specify how many birds were successful on the confirmation trial.

Response: We added how many birds solved the problem a second time for each problem-solving task [L585-586, 590-592, 638-639, 644-645].

L569: should this say 'day 2' rather than 'day 1' to match L552?

Response: Yes, we corrected this.

L595: if a bird chose the rewarded colour in trials 1-7, would they be given a score of 1, 7, or 3.5. Should clarify scoring scheme.

Response: This part now reads: "The success criterion for associative learning was seven consecutive correct trials excluding the training trials; thus, the best possible score was 7 trials" [L624-625].

L618: typo 'residues'; also, is the relationship between # innovations and research effort necessarily linear? I would think that # of innovations would plateau with increasing effort, though perhaps few species have sufficient research effort to reach that point.

Response: We corrected "residues" to "residuals". Those were computed using the whole innovation database. We plotted a graph of the innovation rates and research effort values from the innovation database below for the reviewer. The relationships between innovation rates and research effort do not appear to deviate significantly from linearity (see Figure below).

It's a common practice to correct innovation values using research effort, as Overington et al. (2009) explain:

source, provide a link to the Creative Commons license, and indicate if changes were made. In the cases where the authors are anonymous, such as is the case for the reports of anonymous peer reviewers, author attribution should be to 'Anonymous Referee' followed by a clear attribution to the source work. The images or other third party material in this file are included in the article's Creative Commons license, unless indicated otherwise in a credit line to the material. If material is not included in the article's Creative Commons license and your intended use is not permitted by statutory regulation or exceeds the permitted use, you will need to obtain permission directly from the copyright holder. To view a copy of this license, visit <http://creativecommons.org/licenses/by/4.0/>.

"To correct our measures for the fact that more intensely studied species inevitably have more innovation reports, we regressed each of the log-transformed innovation measures against log-transformed research effort. Research effort is defined as the number of scientific papers published on a given taxon according to Zoological Records' web index (1978–2004). This index covers all the types of journals from which our innovation database is collated. There is a strong relationship between the frequency of innovation reports and research effort in our data set ($R^2 = 0.75$). Previous work (Nicolakakis & Lefebvre 2000; Morand-Ferron et al. 2007) has also shown that research effort is highly correlated with species number per taxon and taxonomic distribution of photos in birding magazines ($R^2 = 0.688$ – 0.889). Because of these correlations, regressing innovation frequency or diversity against research effort also controls for speciosity and differential interest by birdwatchers. We used the Studentised residuals of the innovation-research effort regressions as the predictor variables in the models with residual brain size as the response variable."

For clarity, we added a sentence in the methods: "as the probability of observing an innovation increases with the time spent observing a species", and cited Overington et al. (2009) [L654-655].

L622: where did the body mass data come from, and were they specific to males given that most birds in the study were male?

Response: We used the body mass from the same database as the brain sizes. We added this information in the methods [L658]. The data is the average of both sexes (when available, see Sayol et al. 2018); therefore, we do not have values specific for males. This is now also mentioned in the methods [L658]. If we were able to separate by sex, the relationship would likely be even stronger due to reducing one variable.

L643-645: should specify how many categories there were for 'capture site,' since I thought all wild birds were captured at the same site (Rockefeller Field Station); also, why was shyness included in the models for neophobia, but neophobia was not included in the models for shyness (extended Table 3); this sentence should also include 'dietary generalism', and another sentence should explain where those data came from

Response: There were 8 capture sites within the ~1,200 acres of the Rockefeller Field Center. We now write this out more clearly in the sentence we had in the methods: "Birds were captured using mist nets placed in 8 sites within a 30-hectare radius around the ~1,200 acres of the Rockefeller University Field Research Center. These sites included four open fields and four forests (distance between each capture site: 200 to 500 meters)" [Lines 441-443].

We added neophobia in the shyness models (Extended Data Table 5, models 1-5)

We added "dietary generalism" with the reference [L700].

L645: typo - change MGMG to MCMC

Response: We corrected the typo.

L664: unclear what is meant by 'excluding seasonal migrants'. Aren't several of your species migratory at your study site (e.g., goldfinch, cowbird, sparrows, phoebe, catbird, wrens)?

Response: We agree with the reviewer that this formulation was incorrect. We changed it to: "excluding migrants that were only passing by" [L736-737].

Extended Fig. 2 caption: tasks are described as 'lid-pulling' and 'lid-flipping', but as 'lid-knocking' and 'lid-flipping' elsewhere

Response: We replaced the mislabelled "lid-knocking" with "lid-pulling".

Extended Fig. 3: fig caption uses directional language, such as 'species that perform better on problem-solving tasks are not shyer or more neophobic, but the basis of those directional predictions is unclear. For example, why couldn't birds with better problem solving be bolder rather than shyer? If the directionality is to be maintained, rationale should be provided, and the correlation tests should be 1-tailed. I recommend changing the wording here to say that trait X was not related to trait Y.

Response: We changed the wording of this legend to reflect the non-directional interpretation.

Extended Table 1: please add a column for family to help readers from outside ornithology

Response: We added families in Extended Table 1.

Extended Table 3: in model 25a, the line for 'reward' should be in bold

Response: Following the updated analyses, this variable is not significant anymore after false discovery correction.

Videos: very helpful, but the versions I had for review were quite grainy; the two showing the self-control test, in particular, were difficult to see what was happening; if higher-quality versions exist, they would be helpful

Response: We re-encoded the videos; they should appear better now (as long as they are not viewed full-screen – this is the best resolution we have for the videos).

Reviewer #3 (Remarks to the Author):

The study uses an impressive range of cognitive tests comparing over 200 individuals across 15 songbird species. This gives depth to the findings regarding a more general concept of factors affecting innovation rates. The study is well executed and written. However, I have some concerns about the calculation of 'pure neophobia', which should be revisited. Moreover, I wondered whether any corrections were used for the many Spearman correlations. If not, this should be added.

Main:

Relevant background information has been provided that leads to the aims. The approach is innovative and one of very few addressing all three steps usually linked to innovation across a good range of songbird species. The paper is relevant for psychologists, behavioural ecologists and cognitive ecologists, and therefore of interest across disciplinary areas.

Recommendation:

Key results: The study investigated several cognitive abilities often linked to innovation using an extensive set of tests in a large sample covering 15 songbird species to establish the underlying factors affecting innovations in the wild. The key finding is that problem-solving abilities in captivity seem to be the best predictor of innovation rates in the wild also correlating with brain size, whereas other factors showed little relationship to innovation rate or brain size. Given the breadth of experiments conducted and the number of individuals and species tested, this constitutes a major step forward in explaining innovation rates in birds.

Validity: Results are well supported by large sample sizes and thorough testing. However, I am concerned that the 'pure neophobia' variable is not representative of neophobia as it is currently calculated (see comment below regarding lines 521-522 and 529-530). Moreover, I could nowhere find that there were any corrections used for multiple testing (Spearman correlations; see comment regarding e.g., lines 633-637). Finally, the authors may consider testing individuals nested within species (see comment regarding lines 643-645). All issues might affect the outcome but can be addressed in a revision.

Response: We thank the reviewer for the positive comments and agree with the reviewer's assessment of our study. Following a similar concern by reviewer 1, we now provide corrected p-values using false discovery rate (FDR) by applying Benjamini & Hochberg p-value correction for all statistical tests (Spearman correlations, MCMCglmm with species means, final MCMCglmm with individual variation in Table 1). While the p-values differ, most conclusions remain, except for the relationship between problem-solving and total innovation when correcting for phylogenetic relationships. We now mention this new result in the manuscript [L129-133]. The adjustment procedure has been added in the methods [L716-722].

Originality and significance: Conclusions are original and provide an important contribution to increase knowledge about factors affecting innovation. The large range of tests used in combination with a large sample size covering 15 songbird species makes this a robust and significant finding. The design is likely to be adopted by other researchers to test similar relationships in other taxa and is therefore a large relevance. The study combined psychological approaches with ecological theory and is of relevance to a broad readership across disciplines.

Data & methodology: Data belong to over 200 individuals of 15 songbird species which is an impressive sample in this field of research. The methodology is thorough and uses an extensive set of tests that surpasses most other studies. Therefore, the data are of high quality. Methods are clearly described and established procedure have been used (but see comments above regarding neophobia).

Appropriate use of statistics and treatment of uncertainties: Appropriate tests have been used (but see comment about corrections for multiple testing). Legends need in part more information (see comments regarding Extended Data Fig. 2 and 10 and main Figs. 3 and 4).

Conclusions: Conclusions are valid and robust and reflect the results.

Suggested improvements:

Fig. 1: The figure provides an informative overview about factors affecting innovations and tests used. However, other factors have been ignored. For example, while dietary generalism has been linked to innovations, persistence has also been identified (Griffin & Guez 2014, Behavioural Processes, 109:121–134. doi:10.1016/j.beproc.2014.08.027.) and should be included here. Moreover, shyness and neophobia have been listed to potentially affect problem-solving abilities. One important factor is missing here as exploration (neophilia) has been repeatedly linked to problem-solving abilities (e.g., Griffin & Guez 2014). It should be mentioned that neophobia and neophilia are not the extremes along a continuum but represent independent motivations (see 2-Factor model in Greenberg & Mettke-Hofmann 2002, Ethology 108, 249-272). This should be added in the text.

Response: We only included in Fig. 1 the traits examined in this study. We changed the title to reflect this: "Links between field measures of innovations, brain size and their potential laboratory measurements, assessed in this study". While persistence and exploration have been identified as factors affecting problem-solving and innovation, we do not have data on persistence or exploration (neophilia) in our tested animals. We do mention that persistence is an important factor affecting innovation, and cite Griffin & Guez 2014 [L72-73]. However, we believe that discussing all other factors not examined in this study would take too much space and may confuse the reader.

One other point: it has nowhere been explained what the solid and hatched lines mean.

Response: We added a description of the regression solid and hashed lines in the legends: "lines were computed using linear regressions on the same variables to illustrate the strength of relationships; solid lines: $P < 0.05$, dashed lines: $P > 0.05$ ". To improve clarity of the graphs (especially since we added species codes following reviewer 2's suggestion), we removed the grey lines that referred to tests excluding the suboscine and domesticated species. We instead present the results without those species in Ext. data tables 2-3.

Results:

Extended Data Fig. 2 and main Figs. 3 and 4: *, ** and *** are not explained in the legend. R and P values are from Spearman correlations. This required many tests. Were there any corrections included, e.g., sequential Bonferroni tests to account for the number of tests performed? If so, this should be mentioned in the legend and the methods section. If no corrections have been done, this would be an important aspect to add to reduce type 1 errors (this applies to all figures and analyses).

Response: In-graph stars referred to the significance of correlations. We kept the regression lines, but we removed the stars to avoid confusion between regression lines and our computed p-values from correlations, to address reviewer 1's concern. To account for multiple testing, we applied Benjamini & Hochberg false discovery rate correction; less conservative than Bonferroni correction (which would likely generate false negatives) while being effective at reducing the risk of false positives. We believe the correction we used represents a good trade-off for our type of analysis.

Extended Data Fig. 10: The y-axis label is missing.

Response: We added the y-axis label.

Extended Data Table 3: While the text mentions that interactions were considered, there is not a single interaction listed in the table. This is odd as it is difficult to imagine that there are no significant interaction terms. Can this please be confirmed, or meaningful interactions be added in the model.

Response: We apologise for the confusion; no interactions were considered in the manuscript. While we agree that interaction effects can be crucial in some contexts, our purpose for including fixed predictors in addition to our variables of interest in the MCMCglmm full modelling was to assess the effect of potential covariates rather than explore how each individual variable covaried. We had no hypothesis on any interaction in the potential covariates we tested, and we couldn't think of meaningful interactions that could explain the relationships we discovered in this study. For the reviewer, we verified the existence of an interaction between shyness and neophobia (as the reviewer suggested below), and this effect was not significant in any of the tested models (all $P > 0.196$).

Lines 143-146: I would not say that some of the relationships were weak associations as problem-solving and shyness for example were consistently highly significant. On what basis do you judge a relationship weak or strong?

Response: We agree with the reviewer and removed "weakly" from this sentence.

Discussion:

Lines 148pp: While I agree that most other cognitive traits were not related to innovation rate or brain size, I think the consistent link between problem-solving ability and shyness needs mentioning and discussion.

Response: Actually, unlike the links between problem solving and innovation and brain size, the link between problem-solving and shyness was not consistent. The later relationship only showed up in the full MCMCglmm analyses. Other relationships that appeared only in the full MCMCglmm analyses included: domesticated species were less shy than wild-caught species; neophobia was negatively associated with associative learning; and food reward type was associated with reversal learning. Given that the focus of our paper is on cognitive assays, not the potential personality confounds, we believe that whatever fine grain patterns we may see for shyness (likely only in some species) goes beyond the scope of our study.

Lines 513-517: For shyness, only the first latency to feed without anything next to the food was measured on day one. Shyness is a personality measure, and an important characteristic of personality is its consistency over time (e.g., Wilson et al. 1994; TREE 9(11), 442-446). However, this consistency has nowhere been shown. The authors report a decline in response times over time (habituation). They could use the same data to test for consistency in responses on the individual level, i.e., are individuals that were slow to feed on the first day still slow to feed on the e.g., last day of testing relative to the other birds. Otherwise, this measure should not be called shyness, particularly as the used measure is quite unusual.

Response: We acknowledge that personality traits should be repeatable. We calculated repeatability for shyness (and neophobia) and found that they were highly repeatable (shyness: $R = 0.440$, $P = 2.04 \times 10^{-41}$, neophobia: $R = 0.301$, $P = 1.71 \times 10^{-20}$). We added the results in Extended Data Table 7 and the manuscript [L548-551], and cited Wilson et al 1994.

Lines 521-522: Why did you set neophobia to zero when the shyness measure was higher than the neophobia measure? The correct way would be to use the correct negative value. Martin II & Fitzgerald (2005; Behavioral Ecology, doi:10.1093/beheco/ari044) for example reported a faster approach to food with novel objects present and interpreted this as attraction to novelty (neophilia). I suggest repeating the analyses with the correct values.

Response: We modified our neophobia measurement per the reviewer's suggestion: neophobia measures now include negative values. We use this new measure throughout the analyses.

Lines 529-530: I wonder whether the average across four days of neophobia testing is a good idea. Your control variable 'shyness' decreased over the four testing days, whereas the time to feed with the novel objects might have stayed the same (one would not expect habituation of neophobia towards different objects). This would increase pure neophobic responses from day 1 to 4 due to the decrease in shyness (you can test for this and when this is indeed the case your pure neophobia measure using the average across the four days is not a true reflection of an individual/species' neophobia and should not be used). This reflects one of the problems using shyness as the control as it is not a consistent representation how fast an individual usually feeds day in and day out. A better way would have been to give the birds more time to habituate and wait until they consistently feed without objects around. This value could then have served as a control measure for neophobia. At the moment, I cannot see how to solve this issue. The first pure neophobia measure is unlikely the correct neophobia response and likely the lowest neophobia among the four measures due to the high shyness measure, i.e., shyness and neophobia are likely to interact here. The last pure neophobia measurement is possibly the best representation of neophobia as the birds have habituated to the experiment and shyness does not interact with neophobia anymore. Maybe just using this last measure would be a better representation of neophobia than the average across the four days unless you can show that pure neophobia responses were highly correlated within individuals across the four days. Another option could be to use the pure neophobia measure of each day for the respective cognitive test, but I assume this would be very difficult to handle. In any way, one or the other alternative should be explored and tested to see whether it has any effect on the results. I would expect that any potential effect of neophobia on problem-solving and/or innovation would increase.

Response: We appreciate the reviewer's careful thoughts about the best possible variables to use in our models. Our strategy is based on the assumption that a bird's shyness should be relatively consistent across the 3 personality trials on any particular day (1st: shyness, 2nd: shyness+neophobia, 3rd: shyness), tested one after the other. If a bird's measured shyness is high for the two shyness trials, it should also be high for the in-between (shyness+neophobia) trial. Therefore, removing this shyness component leaves the remaining latency as "pure neophobia", with minimal, if at all, impact from the shyness measure (as long as the neophobia values are not too often capped – which happened only for 22/801 neophobia trials – 2.7% of the cases – in our dataset). Our data do not indicate that shyness values have an impact on our neophobia measures. Tests of relationships between shyness and neophobia show that they are not related (see Extended Table 5, models 1-10). Eliminating all other variables and assessing only the relationship between shyness and neophobia yield the same result:

Shyness (all days) ~ Neophobia (all days) : post.mean: -0.115 [-0.254, 0.018], P_{MCMC} = 0.0955

In addition, as stated above, shyness and neophobia do not interact. Importantly, our neophobia measures are significantly repeatable across the four measurements [L560-562].

For the reviewer, we retested the models assessing the potential effect of neophobia on the relationship between problem-solving and innovation using the neophobia calculated only on day 4. Neophobia on day 4 did not have more effect than the average neophobia calculated with all four measurement days.

Problem-Solving \sim *Technical innovation* + *Neophobia (day 4)*:

Neophobia post-mean: 0.016 [-0.021, 0.062], $P_{MCMC} = 0.4635$ (*innovation*: NS)

Problem-Solving \sim *Technical innovation* + *Neophobia (mean all days)*:

Neophobia post-mean: -0.114 [-0.253, 0.026], $P_{MCMC} = 0.1171$ (*innovation*: NS)

Finally, we cannot rule out the possibility that some individuals or species are more fearful of objects of specific shapes or colours; thus, using different objects to measure neophobia is likely a more accurate assessment overall. Therefore, keeping the mean of 4 neophobia and 4 shyness measurements better reflects the overall neophobia and shyness for each individual and species. Moreover, as the reviewer pointed out, consistency in personality measurements over time is crucial. Hence, using our 4 measures of neophobia (which we now show is highly repeatable) is coherent with this idea, in addition to being more consistent with our approach for shyness measures.

Lines 633-637: Many Spearman correlations were used resulting in type 1 errors. Were any corrections used (e.g., sequential Bonferroni adjustments) to account for this? If not, this should be done.

Response: P-values from all statistical analyses have been adjusted for multiple testing.

Lines 643-645: Species were included as random factors. This accounts for repeated testing of the same species. However, shouldn't there be a nested approach be used with testing individuals within species to account for closely related individuals (e.g., belonging to the same species) to be more similar in their responses?

Response: We included in models both a "species" random factor to account for testing multiple individuals per species, and a phylogenetic random factor "Phylogeny" (see methods L695-699, available code, and Table outputs).

References: Relevant references are included.

Clarity and context: Abstract and main text are well written.

Decision Letter, first revision:

22nd August 2023

Dear Dr Audet,

Your manuscript entitled "Problem-solving skills are predicted by innovations in the wild and brain size in Passerines" has now been seen by 2 reviewers, whose comments are attached. Unfortunately, Reviewer 3 was unavailable to re-review the manuscript due to other commitments, and their comments and your responses to them have been assessed by Reviewer 2. Both reviewers (1 and 2) have now raised some concerns which will need to be addressed before we can offer publication in Nature Ecology & Evolution. As such, we will need to see your responses to their comments along with a revised manuscript, before we can reach a final decision regarding publication.

Please revise your manuscript taking into account all reviewer comments and highlight all changes in the manuscript text file. We will send the revised manuscript back for another round of review with the same referees.

As before, please take the following into account when revising your manuscript:

* If you have not done so already please begin to revise your manuscript so that it conforms to our Article format instructions at <http://www.nature.com/natecolevol/info/final-submission>. Refer also to any guidelines provided in this letter.

[REDACTED]

Note: This URL links to your confidential home page and associated information

23about manuscripts you may have submitted, or that you are reviewing for us. If you wish to forward this email to co-authors, please delete the link to your homepage.

We hope to receive your revised manuscript within two to four weeks. If you cannot send it within this time, please let us know. We will be happy to consider your revision so long as nothing similar has been accepted for publication at Nature Ecology & Evolution or published elsewhere.

Nature Ecology & Evolution is committed to improving transparency in authorship. As part of our efforts in this direction, we are now requesting that all authors identified as 'corresponding author' on published papers create and link their Open Researcher and Contributor Identifier (ORCID) with their account on the Manuscript Tracking System (MTS), prior to acceptance. ORCID helps the scientific community achieve unambiguous attribution of all scholarly contributions. You can create and link your ORCID from the home page of the MTS by clicking on 'Modify my Springer Nature account'. For more information please visit www.springernature.com/orcid.

[REDACTED]

Reviewers' comments:

Reviewer #1 (Remarks to the Author):

The authors did a very good job revising the manuscript. Now it is much easier to understand the applied analyses and their rationales. It is also reassuring that basically the same results remained after applying phylogenetic control and p value adjustment all over the analyses.

I would, however, strongly suggest further simplifications. Please remove all rank correlation analyses from the whole MS. After that you have proper phylogenetic control you do not need these correlations. They just complicate the understanding of this excellent MS. Please report only the results of the MCMCglimm analyses. If you want to retain the regression lines in the figures now you can do it correctly by using the parameter estimates from MCMCglimm runs.

I would also suggest to remove the finding that total innovation is related to problem solving. This is only supported by the rank correlations but it disappears after proper phylogenetic control, therefore its meaning is dubious. Removing this part simplifies the MS further, you do not need to explain why you have different analyses and why the different analyses provides different results. You can concentrate on your main message strongly supported by a correct analysis.

24Some small issues:

- Table 1: it is unclear how the p values were adjusted in this table. Please explain.
- Part of the legend of Fig 4 is duplicated (L. 429-431 vs L. 420-422).

Reviewer #2 (Remarks to the Author):

I have reviewed the revised manuscript and the authors' responses to the reviewers. The authors thoroughly addressed all of my previous comments. In my opinion, they also fully addressed the comments raised by the other reviewers, with one possible exception:

Reviewer 3 requested that the authors calculate and report repeatability coefficients for shyness and neophobia, since these are described in the manuscript as personality traits that, by definition, must be repeatable. The authors complied and reported repeatability in the text and in extended data table 7. However, repeatability was calculated using scores from 203 individuals across 15 species, but without including species as a factor in the analysis. Therefore, I believe the high levels of repeatability may reflect consistent differences among species rather than among individuals within species. I recommend expanding extended data table 7 to show repeatability per species, or re-run the current analysis but including species as a factor so that repeatability estimates are based on consistent inter-individual differences within species.

Otherwise, I believe the manuscript is ready for publication and congratulate the authors on a very interesting manuscript.

Sincerely,
Dave Wilson

*****END*****

Author Rebuttal, first revision:Reviewers' comments:

Reviewer #1:

The authors did a very good job revising the manuscript. Now it is much easier to understand the applied analyses and their rationales. It is also reassuring that basically the same results remained after applying phylogenetic control and p value adjustment all over the analyses.

Response: We thank the reviewer for taking the time to review our manuscript once again. We also think that it is strengthened by the addition of the reviewer's suggestions.

I would, however, strongly suggest further simplifications. Please remove all rank correlation analyses from the whole MS. After that you have proper phylogenetic control you do not need these correlations. They just complicate the understanding of this excellent MS. Please report only the results of the MCMCgIimm analyses. If you want to retain the regression lines in the figures now you can do it correctly by using the parameter estimates from MCMCgIimm runs.

Response: We could in theory remove all correlation analyses, which could simplify the manuscript. However, we think that showing correlation analyses of the raw data first, followed by more conservative, independent statistical tests that include phylogenetic and corrections for multiple comparisons, makes a coherent and more comprehensive story. It provides all the necessary evidence to let the reader appreciate the full scope of the results. Other studies often don't apply the sophisticated corrections we conducted in the later stages of our paper.

I would also suggest to remove the finding that total innovation is related to problem solving. This is only supported by the rank correlations but it disappears after proper phylogenetic control, therefore its meaning is dubious. Removing this part simplifies the MS further, you do not need to explain why you have different analyses and why the different analyses provides different results. You can concentrate on your main message strongly supported by a correct analysis.

Response: we have added the word "technical" to the term "feeding innovations" in the abstract to reflect this detail of our re-analyses. In the current version, we present the results of innovation in the following way: *"In line with this prediction, we detected no significant association between problem-solving performance and food-type innovation, but we did find a strong correlation with technical innovation rates (...) The above relationships remained significant when controlling for phylogeny and Benjamini-Hochberg false discovery rate (FDR) corrections, except for total innovation (food and technical combined), which fell short of traditional significance levels ($P = 0.0813$) when controlling for phylogeny."*

Some small issues:

- Table 1: it is unclear how the p values were adjusted in this table. Please explain.

Response: We used the Benjamini-Hochberg false discovery rate (FDR) correction using the "p.adjust" function in R, "BH" method (methods, L716-718). The corrections were applied per group of repeated tests (e.g., 5 models testing innovation and brain size effects on each behavioural measurement). We now mention this also in the main text.

- Part of the legend of Fig 4 is duplicated (L. 429-431 vs L. 420-422).

Response: We thank the reviewer for spotting that mistake; we corrected it.

Reviewer #2 (Remarks to the Author):

I have reviewed the revised manuscript and the authors' responses to the reviewers. The authors thoroughly addressed all of my previous comments. In my opinion, they also fully addressed the comments raised by the other reviewers, with one possible exception:

Reviewer 3 requested that the authors calculate and report repeatability coefficients for shyness and neophobia, since these are described in the manuscript as personality traits that, by definition, must be repeatable. The authors complied and reported repeatability in the text and in extended data table 7. However, repeatability was calculated using scores from 203 individuals across 15 species, but without including species as a factor in the analysis. Therefore, I believe the high levels of repeatability may reflect consistent differences among species rather than among individuals within species. I recommend expanding extended data table 7 to show repeatability per species, or re-run the current analysis but including species as a factor so that repeatability estimates are based on consistent inter-individual differences within species.

Response: We agree that not considering species as a factor in our model may not have eliminated species as a repeatability effect on the personality measurements. We reran the repeatability model with the species included as a fixed effect. The repeatability values are in fact now lower, but still considerable and statistically significant. None of our conclusions have changed concerning repeatability of shyness and neophobia. We have updated the manuscript and table S7 with the new values.

Otherwise, I believe the manuscript is ready for publication and congratulate the authors on a very interesting manuscript.

Response: We appreciate the thorough review of our manuscript, and thank the reviewer for the positive evaluation of our paper.

source, provide a link to the Creative Commons license, and indicate if changes were made. In the cases where the authors are anonymous, such as is the case for the reports of anonymous peer reviewers, author attribution should be to 'Anonymous Referee' followed by a clear attribution to the source work. The images or other third party material in this file are included in the article's Creative Commons license, unless indicated otherwise in a credit line to the material. If material is not included in the article's Creative Commons license and your intended use is not permitted by statutory regulation or exceeds the permitted use, you will need to obtain permission directly from the copyright holder. To view a copy of this license, visit <http://creativecommons.org/licenses/by/4.0/>.

Reviewer #1

The authors did a good job to revise the MS. On the other hand, I still insist to remove the correlations on the raw data.

First, you don't need them.

Second, it is not elegant to have them. You have correct analyses with phylogenetic control and p value adjustment, so why to complicate the paper unnecessarily. Furthermore, you can avoid statements like "which fell short of traditional significance level" which can be read in a way that you are not happy with this result.

Third, they result in inconsistencies. For instance, you wrote on line 162 that "In contrast, the negative association between self-control and food innovation and absolute brain size were no longer significant..." but this information is already given on lines 139-141, i.e. it is not "in contrast". Furthermore, in the caption of Figure 4 you state that "problem-solving performance across species is associated with their total innovation reports" but on page 5 you just state the opposite that there is no relationship between these two after controlling for phylogeny. Here, you also mention that tests were done with phylogeny correction and p value adjustments but, showing some level of ignorance, fail to state that some of the results changed as a consequence of using these well justified statistical improvements.

Fourth, I still maintain that it is not correct to draw trend lines on scatter plots showing correlations. The two statistical methods have rather distinct assumptions, for instance (i) for regression you have to distinguish between explanatory and response variables while for correlation you do not and (ii) you have to know the explanatory values rather accurately for a regression model which is not a requirement for correlations (just to name a few). Also, having this trend lines from linear regression is a kind of superfluous here because you have the proper phylogenetically controlled regression which can be used here to draw the trend lines.

I admit that the above points might be seen as a kind of different taste, but they are not. Nevertheless, I leave the decision for the editors of this prestigious journal whether they allow the paper to appear as it is or require the changes I suggested above.

Response: We thank the reviewer for evaluating our manuscript once again. Following their and the editor's suggestion, we removed all correlation tests from the manuscript, which includes tables, figures,

29and text. Because food and technical innovations analyses were already presented separately, we removed the combined variable “total innovations”, not significant with phylogenetic tests, to eliminate the redundancy. In addition, the tables previously presented most analyses in 3 ways: all species, songbirds only, and wild birds only. Following the elimination of correlation data, this distinction is not indicated anymore since all analyses now include variables of phylogeny and captive status, which take into account those differences. Overall, this simplifies the manuscript, and we thank the reviewer for their suggestions.

Finally, trend lines on graphs now represent predicted values from phylogenetic models (using estimate and slope outputs from phylogenetic MCMCglmm). This is made clear throughout the legends.

L201: "vocal learning birds and primates" -> "vocal learning in birds and primates"?

Response: We agree that this wording was confusing. We removed “vocal learning”, which was unnecessary in this sentence.

Reviewer #2:

The authors have addressed all of my concerns and I believe the manuscript should now be published. I congratulate the authors on a fascinating study. Sincerely, Dave Wilson

Response: We thank the reviewer for this additional review and the positive comment.

List of changes:

Title: we added “technical”

L101: we added “Phylogenetic Bayesian mixed models (MCMCglmm) conducted...”

L109: we added “using linear mixed models”

L122-124: we removed significant result of correlation with total innovation

L136-140: we removed the text stating the discrepancy between correlation and phylogenetically controlled results since the correlation result is not presented anymore.

L144-149: we removed the correlation result between self-control and food innovation.

L156: we made it clear the this part uses the complete dataset of 203 values as opposed to the interspecific MCMCglmm analyses.

L173-175: we removed the text stating the discrepancy between correlation and phylogenetically controlled results since the correlation result is not presented anymore.

L185: we added “technical” innovation

L215: we removed “vocal learning”

L660-715: we updated the methods of the statistical analyses to reflect the above changes.

We also updated all figures, tables and legends to reflect the new analyses, as suggested by reviewers.

Decision Letter, second revision:

3rd November 2023

Dear Dr. Audet,

Thank you for submitting your revised manuscript "Problem-solving skills are predicted by innovations in the wild and brain size in Passerines" (NATECOLEVOL-23020481B). It has now been seen again by the original reviewers and their comments are below. The reviewers find that the paper has improved in revision, and therefore we'll be happy in principle to publish it in Nature Ecology & Evolution, pending minor revisions to satisfy the reviewers' final requests and to comply with our editorial and formatting guidelines.

In particular, we agree with Reviewer 1's view that the rank correlations do not need to be included in the manuscript, given your inclusion of phylogenetically controlled analyses. Therefore, please prepare a revised version of the manuscript without these analyses and make appropriate changes to the text and figures. I would also request you to prepare another short 'response to referees' file indicating where you have made these changes (page/line numbers and figure legends, as appropriate).

We are now performing detailed checks on your paper and will send you a checklist detailing our editorial and formatting requirements in about a week. Please do not upload the final materials or make any additional changes until you receive this information from us. However, as noted above, please start revising the manuscript according to Reviewer 1's suggestions, as we will need to see

31these changes before finalizing the paper for publication.

[REDACTED]

Reviewer #1 (Remarks to the Author):

Comments for the authors

The authors did a good job to revise the MS. On the other hand, I still insist to remove the correlations on the raw data.

First, you don't need them.

Second, it is not elegant to have them. You have correct analyses with phylogenetic control and p value adjustment, so why to complicate the paper unnecessarily. Furthermore, you can avoid statements like "which fell short of traditional significance level" which can be read in a way that you are not happy with this result.

Third, they result in inconsistencies. For instance, you wrote on line 162 that "In contrast, the negative association between self-control and food innovation and absolute brain size were no longer significant..." but this information is already given on lines 139-141, i.e. it is not "in contrast". Furthermore, in the caption of Figure 4 you state that "problem-solving performance across species is associated with their total innovation reports" but on page 5 you just state the opposite that there is no relationship between these two after controlling for phylogeny. Here, you also mention that tests were done with phylogeny correction and p value adjustments but, showing some level of ignorance, fail to state that some of the results changed as a consequence of using these well justified statistical improvements.

Fourth, I still maintain that it is not correct to draw trend lines on scatter plots showing correlations. The two statistical methods have rather distinct assumptions, for instance (i) for regression you have to distinguish between explanatory and response variables while for correlation you do not and (ii) you have to know the explanatory values rather accurately for a regression model which is not a requirement for correlations (just to name a few). Also, having this trend lines from linear regression is a kind of superfluous here because you have the proper phylogenetically controlled regression which can be used here to draw the trend lines.

I admit that the above points might be seen as a kind of different taste, but they are not. Nevertheless, I leave the decision for the editors of this prestigious journal whether they allow the paper to appear as it is or require the changes I suggested above.

L201: "vocal learning birds and primates" -> "vocal learning in birds and primates"?

32Reviewer #2 (Remarks to the Author):

The authors have addressed all of my concerns and I believe the manuscript should now be published. I congratulate the authors on a fascinating study. Sincerely, Dave Wilson

Our ref: NATECOLEVOL-23020481B

20th December 2023

Dear Dr. Audet,

Thank you for your patience as we've prepared the guidelines for final submission of your Nature Ecology & Evolution manuscript, "Problem-solving skills are predicted by innovations in the wild and brain size in Passerines" (NATECOLEVOL-23020481B). Please carefully follow the step-by-step instructions provided in the attached file, and add a response in each row of the table to indicate the changes that you have made. Please also check and comment on any additional marked-up edits we have proposed within the text. Ensuring that each point is addressed will help to ensure that your revised manuscript can be swiftly handed over to our production team.

****We would like to start working on your revised paper, with all of the requested files and forms, as soon as possible (preferably within two weeks). Please get in contact with us immediately if you anticipate it taking more than two weeks to submit these revised files.****

In recognition of the time and expertise our reviewers provide to Nature Ecology & Evolution's editorial

33process, we would like to formally acknowledge their contribution to the external peer review of your manuscript entitled "Problem-solving skills are predicted by innovations in the wild and brain size in Passerines". For those reviewers who give their assent, we will be publishing their names alongside the published article.

Nature Ecology & Evolution offers a Transparent Peer Review option for new original research manuscripts submitted after December 1st, 2019. As part of this initiative, we encourage our authors to support increased transparency into the peer review process by agreeing to have the reviewer comments, author rebuttal letters, and editorial decision letters published as a Supplementary item. When you submit your final files please clearly state in your cover letter whether or not you would like to participate in this initiative. Please note that failure to state your preference will result in delays in accepting your manuscript for publication.

Cover suggestions

We welcome submissions of artwork for consideration for our cover. For more information, please see our [guide for cover artwork](https://www.nature.com/documents/Nature_covers_author_guide.pdf).

Nature Ecology & Evolution has now transitioned to a unified Rights Collection system which will allow our Author Services team to quickly and easily collect the rights and permissions required to publish your work. Approximately 10 days after your paper is formally accepted, you will receive an email in providing you with a link to complete the grant of rights. If your paper is eligible for Open Access, our Author Services team will also be in touch regarding any additional information that may be required to arrange payment for your article.

Please note that *Nature Ecology & Evolution* is a Transformative Journal (TJ). Authors may publish their research with us through the traditional subscription access route or make their paper immediately open access through payment of an article-processing charge (APC). Authors will not be required to make a final decision about access to their article until it has been accepted. [Find out more about Transformative Journals](https://www.springernature.com/gp/open-research/transformative-journals)

Authors may need to take specific actions to achieve [compliance with funder and institutional open access mandates](https://www.springernature.com/gp/open-research/funding/policy-compliance-faqs). If your research is supported by a funder that requires immediate open access (e.g. according to [Plan S principles](https://www.springernature.com/gp/open-research/plan-s-compliance))

34then you should select the gold OA route, and we will direct you to the compliant route where possible. For authors selecting the subscription publication route, the journal's standard licensing terms will need to be accepted, including <https://www.nature.com/nature-portfolio/editorial-policies/self-archiving-and-license-to-publish>. Those licensing terms will supersede any other terms that the author or any third party may assert apply to any version of the manuscript.

For information regarding our different publishing models please see our <https://www.springernature.com/gp/open-research/transformative-journals> Transformative Journals page. If you have any questions about costs, Open Access requirements, or our legal forms, please contact ASJournals@springernature.com.

[REDACTED]

[REDACTED]

Reviewer #1:

Remarks to the Author:

Comments for the authors

The authors did a good job to revise the MS. On the other hand, I still insist to remove the correlations on the raw data.

First, you don't need them.

Second, it is not elegant to have them. You have correct analyses with phylogenetic control and p value adjustment, so why to complicate the paper unnecessarily. Furthermore, you can avoid statements like "which fell short of traditional significance level" which can be read in a way that you are not happy with this result.

Third, they result in inconsistencies. For instance, you wrote on line 162 that "In contrast, the negative association between self-control and food innovation and absolute brain size were no longer significant..." but this information is already given on lines 139-141, i.e. it is not "in contrast". Furthermore, in the caption of Figure 4 you state that "problem-solving performance across species is associated with their total innovation reports" but on page 5 you just state the opposite that there is no relationship between these two after controlling for phylogeny. Here, you also mention that tests were done with phylogeny correction and p value adjustments but, showing some level of ignorance, fail to state that some of the results changed as a consequence of using these well justified statistical improvements.

35Fourth, I still maintain that it is not correct to draw trend lines on scatter plots showing correlations. The two statistical methods have rather distinct assumptions, for instance (i) for regression you have to distinguish between explanatory and response variables while for correlation you do not and (ii) you have to know the explanatory values rather accurately for a regression model which is not a requirement for correlations (just to name a few). Also, having this trend lines from linear regression is a kind of superfluous here because you have the proper phylogenetically controlled regression which can be used here to draw the trend lines.

I admit that the above points might be seen as a kind of different taste, but they are not. Nevertheless, I leave the decision for the editors of this prestigious journal whether they allow the paper to appear as it is or require the changes I suggested above.

L201: "vocal learning birds and primates" -> "vocal learning in birds and primates"?

Reviewer #2:

Remarks to the Author:

The authors have addressed all of my concerns and I believe the manuscript should now be published. I congratulate the authors on a fascinating study. Sincerely, Dave Wilson

Author Rebuttal, first revision:

Reviewer #1

The authors did a good job to revise the MS. On the other hand, I still insist to remove the correlations on the raw data.

First, you don't need them.

Second, it is not elegant to have them. You have correct analyses with phylogenetic control and p value adjustment, so why to complicate the paper unnecessarily. Furthermore, you can avoid statements like "which fell short of traditional significance level" which can be read in a way that you are not happy with this result.

Third, they result in inconsistencies. For instance, you wrote on line 162 that "In contrast, the negative association between self-control and food innovation and absolute brain size were no longer significant..." but this information is already given on lines 139-141, i.e. it is not "in contrast". Furthermore, in the caption of Figure 4 you state that "problem-solving performance across species is associated with their total innovation reports" but on page 5 you just state the opposite that there is no relationship between these two after controlling for phylogeny. Here, you also mention that tests were

36done with phylogeny correction and p value adjustments but, showing some level of ignorance, fail to state that some of the results changed as a consequence of using these well justified statistical improvements.

Fourth, I still maintain that it is not correct to draw trend lines on scatter plots showing correlations. The two statistical methods have rather distinct assumptions, for instance (i) for regression you have to distinguish between explanatory and response variables while for correlation you do not and (ii) you have to know the explanatory values rather accurately for a regression model which is not a requirement for correlations (just to name a few). Also, having this trend lines from linear regression is a kind of superfluous here because you have the proper phylogenetically controlled regression which can be used here to draw the trend lines.

I admit that the above points might be seen as a kind of different taste, but they are not. Nevertheless, I leave the decision for the editors of this prestigious journal whether they allow the paper to appear as it is or require the changes I suggested above.

Response: We thank the reviewer for evaluating our manuscript once again. Following their and the editor's suggestion, we removed all correlation tests from the manuscript, which includes tables, figures, and text. Because food and technical innovations analyses were already presented separately, we removed the combined variable "total innovations", not significant with phylogenetic tests, to eliminate the redundancy. In addition, the tables previously presented most analyses in 3 ways: all species, songbirds only, and wild birds only. Following the elimination of correlation data, this distinction is not indicated anymore since all analyses now include variables of phylogeny and captive status, which take into account those differences. Overall, this simplifies the manuscript, and we thank the reviewer for their suggestions.

Finally, trend lines on graphs now represent predicted values from phylogenetic models (using estimate and slope outputs from phylogenetic MCMCglmm). This is made clear throughout the legends.

L201: "vocal learning birds and primates" -> "vocal learning in birds and primates"?

Response: We agree that this wording was confusing. We removed "vocal learning", which was unnecessary in this sentence.

Reviewer #2:

The authors have addressed all of my concerns and I believe the manuscript should now be published. I congratulate the authors on a fascinating study. Sincerely, Dave Wilson

37Response: We thank the reviewer for this additional review and the positive comment.

List of changes:

Title: we added “technical”

L101: we added “Phylogenetic Bayesian mixed models (MCMCglmm) conducted...”

L109: we added “using linear mixed models”

L122-124: we removed significant result of correlation with total innovation

L136-140: we removed the text stating the discrepancy between correlation and phylogenetically controlled results since the correlation result is not presented anymore.

L144-149: we removed the correlation result between self-control and food innovation.

L156: we made it clear the this part uses the complete dataset of 203 values as opposed to the interspecific MCMCglmm analyses.

L173-175: we removed the text stating the discrepancy between correlation and phylogenetically controlled results since the correlation result is not presented anymore.

L185: we added “technical” innovation

L215: we removed “vocal learning”

L660-715: we updated the methods of the statistical analyses to reflect the above changes.

We also updated all figures, tables and legends to reflect the new analyses, as suggested by reviewers.

Final Decision Letter:

23rd January 2024

Dear Dr Audet,

We are pleased to inform you that your Article entitled "Problem-solving skills are predicted by

38technical innovations in the wild and brain size in Passerines", has now been accepted for publication in Nature Ecology & Evolution.

Over the next few weeks, your paper will be copyedited to ensure that it conforms to Nature Ecology and Evolution style. Once your paper is typeset, you will receive an email with a link to choose the appropriate publishing options for your paper and our Author Services team will be in touch regarding any additional information that may be required

Due to the importance of these deadlines, we ask you please us know now whether you will be difficult to contact over the next month. If this is the case, we ask you provide us with the contact information (email, phone and fax) of someone who will be able to check the proofs on your behalf, and who will be available to address any last-minute problems . Once your paper has been scheduled for online publication, the Nature press office will be in touch to confirm the details.

Acceptance of your manuscript is conditional on all authors' agreement with our publication policies (see www.nature.com/authors/policies/index.html). In particular your manuscript must not be published elsewhere and there must be no announcement of the work to any media outlet until the publication date (the day on which it is uploaded onto our web site).

Please note that *Nature Ecology & Evolution* is a Transformative Journal (TJ). Authors may publish their research with us through the traditional subscription access route or make their paper immediately open access through payment of an article-processing charge (APC). Authors will not be required to make a final decision about access to their article until it has been accepted. [Find out more about Transformative Journals](https://www.springernature.com/gp/open-research/transformative-journals)

Authors may need to take specific actions to achieve [compliance](https://www.springernature.com/gp/open-research/funding/policy-compliance-faqs) with funder and institutional open access mandates. If your research is supported by a funder that requires immediate open access (e.g. according to [Plan S principles](https://www.springernature.com/gp/open-research/plan-s-compliance)) then you should select the gold OA route, and we will direct you to the compliant route where possible. For authors selecting the subscription publication route, the journal's standard licensing terms will need to be accepted, including [self-archiving-and-license-to-publish](https://www.nature.com/nature-portfolio/editorial-policies/self-archiving-and-license-to-publish). Those licensing terms will supersede any other terms that the author or any third party may assert apply to any version of the manuscript.

We welcome the submission of potential cover material (including a short caption of around 40 words) related to your manuscript; suggestions should be sent to Nature Ecology & Evolution as electronic files (the image should be 300 dpi at 210 x 297 mm in either TIFF or JPEG format). Please note that such pictures should be selected more for their aesthetic appeal than for their scientific content, and that colour images work better than black and white or grayscale images. Please do not try to design a cover with the Nature Ecology & Evolution logo etc., and please do not submit composites of images related to your work. I am sure you will understand that we cannot make any promise as to whether any of your suggestions might be selected for the cover of the journal.

You can generate the link yourself when you receive your article DOI by entering it here: <http://authors.springernature.com/share>.

[REDACTED]

P.S. Click on the following link if you would like to recommend Nature Ecology & Evolution to your librarian <http://www.nature.com/subscriptions/recommend.html#forms>

** Visit the Springer Nature Editorial and Publishing website at http://editorial-jobs.springernature.com?utm_source=ejp_NEcoE_email&utm_medium=ejp_NEcoE_email&utm_campaign=ejp_NEcoE for more information about our career opportunities. If you have any questions please click [here](mailto:editorial.publishing.jobs@springernature.com).**